# Melatonin delays ovarian aging in mice by slowing down the exhaustion of ovarian reserve

Chan Yang[1,2,3,5], Qinghua Liu[1,2,3,5], Yingjun Chen[1,2,3], Xiaodong Wang[1,2,3], Zaohong Ran[1,2,3], Fang Fang[1,2,3], Jiajun Xiong[1,2,3], Guoshi Liu[4], Xiang Li[1,2,3], Liguo Yang[1,2,3] & Changjiu He [1,2,3✉]

Studies have shown that melatonin (MLT) can delay ovarian aging, but the mechanism has not been fully elucidated. Here we show that granulosa cells isolated from mice follicles can synthesize MLT; the addition of MLT in ovary culture system inhibited follicle activation and growth; In vivo experiments indicated that injections of MLT to mice during the follicle activation phase can reduce the number of activated follicles by inhibiting the PI3K-AKT-FOXO3 pathway; during the early follicle growth phase, MLT administration suppressed follicle growth and atresia, and multiple pathways involved in folliculogenesis, including PI3K-AKT, were suppressed; MLT deficiency in mice increased follicle activation and atresia, and eventually accelerated age-related fertility decline; finally, we demonstrated that prolonged high-dose MLT intake had no obvious adverse effect. This study presents more insight into the roles of MLT in reproductive regulation that endogenous MLT delays ovarian aging by inhibiting follicle activation, growth and atresia.

[1] Key Laboratory of Agricultural Animal Genetics, Breeding and Reproduction of Ministry of Education, Huazhong Agricultural University, Wuhan 430070, China. [2] National Center for International Research on Animal Genetics, Breeding and Reproduction, Huazhong Agricultural University, Wuhan 430070, China. [3] College of Animal Science and Technology, Huazhong Agricultural University, Wuhan 430070, China. [4] College of Animal Science and Technology, China Agricultural University, Beijing 100193, China. [5]These authors contributed equally: Chan Yang, Qinghua Liu. ✉email: chungjoe@mail.hzau.edu.cn

The exhaustion of ovarian follicle reserve leads to the loss of ovarian function in middle-aged females, namely ovarian aging[1]. It is generally accepted that ovarian aging result in the loss of reproductive capability and the disorder of reproductive hormones; the latter further trigger many diseases including cardiovascular disease, ovarian cancer, osteoporosis, obesity, and menopausal syndrome[2–5]. Therefore, the impact of ovarian aging reaches far beyond just the reproductive system and it also indirectly causes dysfunction of other organs[6]. With the development of medical technology, the various risks posed by elderly pregnancy can be technically avoided. Therefore, the treatment or delay of ovarian aging is becoming an urgent need for women.

The mainstream view tends to suggest that there are no germline stem cells in the ovary to continuously replenish the ovarian follicle reserve. Therefore, the exhaustion of the ovarian follicle reserve in middle-aged females is the direct cause of ovarian aging[7,8]. Early folliculogenesis, which begins with primordial follicle activation and ends with the formation of small antral follicle (SAF), is directly responsible for the decrease of ovarian follicle reserve[9]. The primordial follicle pool established during fetal or neonatal period is believed to be the pre-established ovarian follicle reserve (PreOR), which maintains the entire reproductive lifespan of the females[10]. During early folliculogenesis, a limited number of primordial follicles are constantly activated from PreOR. The activation rate of follicles is strictly regulated; if not, it will cause premature ovarian failure[11]. Subsequently, the activated follicles develop into primary, secondary, and eventually SAFs, a process also known as early follicle growth[12]. All the SAFs constitute a dynamic ovarian follicle reserve (DOR), the size of which determines how many follicles are selected for cyclic recruitment by gonadotrophin (GTH), or eliminated via atresia.

PreOR and DOR are functionally related and their dynamics are mainly controlled by two pathways, the PI3K-AKT and the BMP/AMH-SMAD[13]. On one hand, DOR can be continuously replenished by the activated follicles from PreOR to maintain its size. Hereby, the number of SAFs in the DOR remain stable before the reproductive ability declines[14], although the number of primordial follicles in the ovary continues to decrease. On the other hand, as the ovary ages, the number of primordial follicles in PreOR greatly decreases, which indirectly leads to reduction in SAFs in DOR. Therefore, the extent of ovarian aging can be reflected by SAF count[15,16]. Of note, SAFs in the DOR will be eliminated via atresia if without sufficient GTH and the number of ovulated eggs per estrous cycle is also fixed. Therefore, the larger the size of DOR, the more SAFs will be eliminated to ensure the stability of fecundity and, accordingly, more primordial follicles will be activated from PreOR to replenish the loss of DOR. Such vicious cycle will eventually result in a meaningless exhaustion of ovarian follicle reserve[13,17,18]. Hence, the vast majority of follicles will be eliminated before puberty, because the hypothalamic pituitary is not mature enough to produce sufficient GTH[19]. In a word, inhibition of early folliculogenesis can directly slow down the exhaustion of ovarian follicle reserve.

Studies have shown that the level of reactive oxygen species (ROS) and inflammatory factors significantly increase in ovary during the function decline period[20,21]. Therefore, researchers have tried some antioxidants to delay ovarian aging such as vitamin C, vitamin E, N-acetyl-L-cysteine, curcumin, coenzyme Q10, proanthocyanidin, quercetin, and resveratrol, and have achieved positive results[22]. As the most representational antioxidant, melatonin (MLT) is also used to delay ovarian aging[23–30]. Studies have demonstrated that shortening the sunshine time can delay reproductive aging, which is considered to be related to MLT[31,32]; long-term administration of MLT in the drinking water of mice can reduce the production of ROS in the ovary, thereby increasing the number of follicles in middle-aged mice[27,29]. MLT administration for rats during the reproductive decline period can help maintain a regular estrous cycle and normal estrogen level, and extend their reproductive age[33]. Clinical study has also found that 6-month MLT administration for perimenopausal and postmenopausal women can decrease the serum GTH level and improve thyroid function[34].

However, there are still several important issues to be elucidated with respect to the research topic of MLT delaying ovarian aging. First, studies have demonstrated that long-term intake of exogenous MLT can effectively delay ovarian aging. However, the question is whether endogenous MLT is involved in the regulation of ovarian aging and, if so, to what extent? Second, in light of the free radical theory of aging, it was generally believed that the antioxidant property of MLT is the pivotal reason for delaying aging[35]. Nevertheless, unlike body aging, the cause of ovarian aging is the exhaustion of the ovarian follicle reserve. Thus, in addition to the antioxidant property, does MLT delay ovarian aging by inhibiting the loss of ovarian follicle reserve? To address these issues, we conducted this study using a mouse model and, finally, demonstrated that endogenous MLT delayed ovarian aging by slowing down the exhaustion of ovarian reserve.

## Results
**MLT may play a regulatory role in early folliculogenesis.** Immunohistochemical data indicated that SNAT, the rate-limiting enzyme in the MLT synthetic pathway, was mainly distributed in the granulosa cells (GCs) (Fig. 1a). Then we examined whether MLT can be synthesized in GCs. To this end, isolated GCs were cultured with or without 5-hydroxytryptamine (5-HT), the precursor of MLT and SNAT substrate (Fig. 1b). High-performance liquid chromatography (HPLC) was used to detect the conversion of 5-HT into MLT. In three cases, namely no addition of GCs and 5-HT (#1–3), the addition of 5-HT but no GCs (#4–6), and the addition of GCs but no 5-HT (#7–9), no MLT was detected in the culture medium. When both GCs and 5-HT were added to the medium (#10–12), MLT absorption peak appeared in the chromatography (Fig. 1c). The above results indicated that follicles have the capacity to synthesize MLT.

Subsequently, we collected ovaries at postnatal day (PD) 5, 7, 9, 11, 13, 15, 17, 19, and 21. The quantitative reverse-transcription PCR (qRT-PCR) data showed that the expression level of *SNAT* in the ovary continued to decrease; *Fshr* and *Lhcgr* are the marker genes that reflected the process of folliculogenesis, and their expression levels gradually increased in the same process, which is opposite to the expression pattern of *SNAT* (Fig. 1d). Next, we examined the MLT level in ovarian homogenates at PD7, 9, 15, 17, and 19. Similar to the expression pattern of *SNAT*, the changes in MLT level gradually decreased with age (Fig. 1e). The above data suggested that MLT may be involved in the regulation of early folliculogenesis.

**MLT addition inhibited early folliculogenesis in in vitro ovary culture system.** To investigate whether MLT is involved in the regulation of primordial follicle activation in in vitro culture system, ovaries from mice at PD3 were cultured in vitro and then the number of activated follicles with and without MLT was counted (Fig. 2a, b). The result showed a significant decrease in the number of activated follicles after both $10^{-8}$ M and $10^{-7}$ M MLT addition ($P = 0.0394$ and $0.025$, respectively). In addition, $10^{-8}$ and $10^{-7}$ M MLT can also significantly reduce the number of atretic follicles ($P = 0.0032$ and $0.0001$, respectively) (Fig. 2c, d).

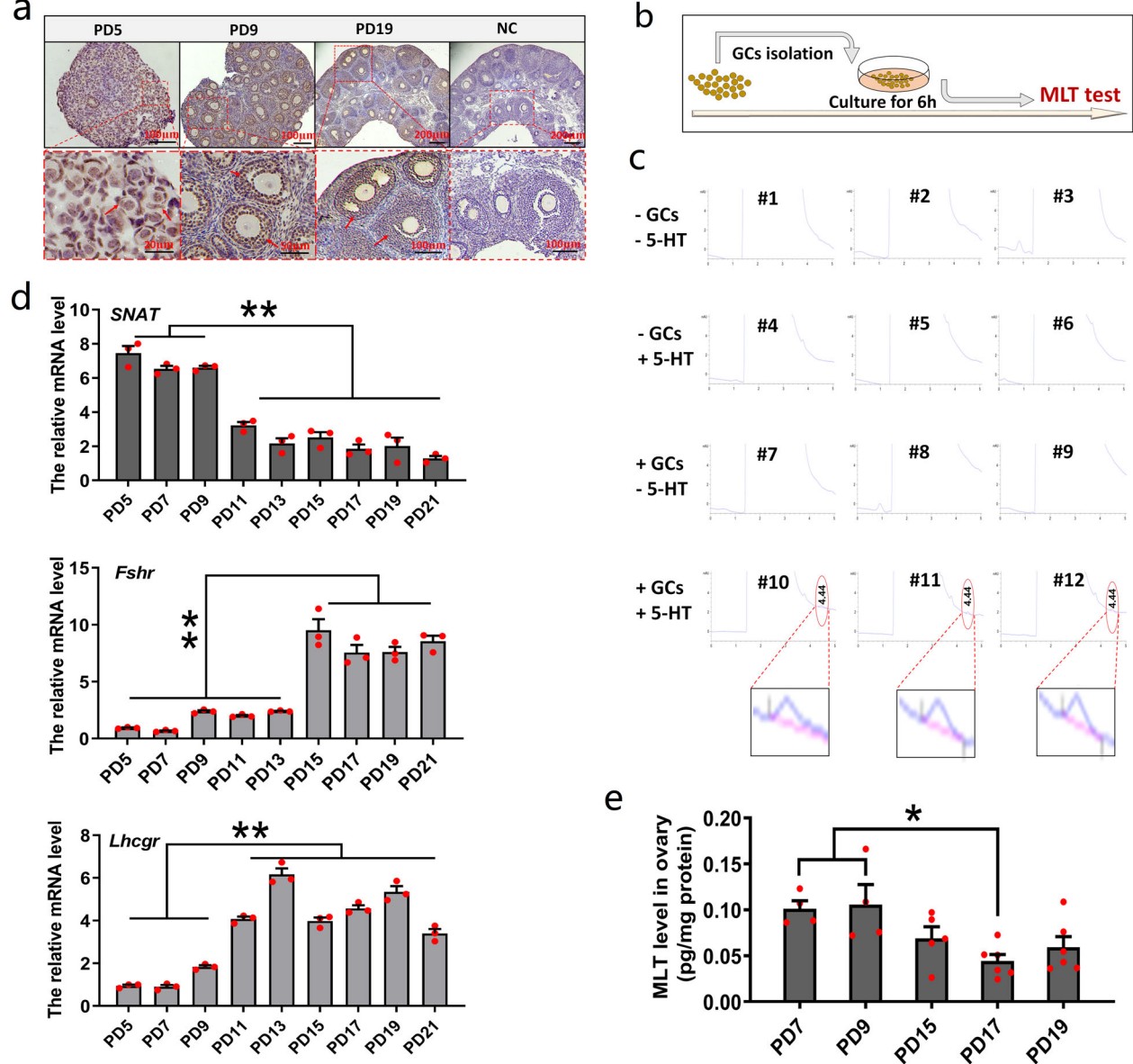

**Fig. 1 MLT may play a regulatory role in early folliculogenesis. a** Location of SNAT in prepubertal ovaries was determined by immunohistochemistry. NC represents negative control. **b**, **c** Detection of the MLT-synthetic ability of GCs. **b** Experimental design. **c** Representative chromatograms showing metabolites extracted from culture medium alone (#1–3), culture medium with 5-HT but no GCs (#4–6), culture medium with GCs but no 5-HT (#7–9), and culture medium with GCs and 5-HT (#10–12). Absorption peaks of MLT only appear in GCs incubated with 5-HT (#10–12, retention time 4.44 min). **d** The expression profiles of *SNAT*, *Fshr*, and *Lhcgr* in prepubertal ovaries were determined by qPCR (values are mean ± SEM). Normalization was performed using the housekeeping gene *Actb*. $n = 3$ biologically independent samples. **e** MLT levels in ovary homogenate (values are mean ± SEM). $n = 4$ (PD7, 9), 5 (PD15), and 6 (PD17, 19) biologically independent samples, respectively. Statistical significance was determined using one-way ANOVA followed by Tukey's post hoc test. Significant differences are denoted by *$P < 0.05$ and **$P < 0.01$.

To further study the effect of MLT on early follicle growth, ovaries from mice at PD10 were isolated and cultured in vitro (Fig. 2e). Then the number of type 5a follicles in the ovaries with and without MLT addition was counted, respectively. It was found that both $10^{-8}$ M and $10^{-7}$ M MLT can significantly reduce the number of type 5a follicles ($P = 0.0348$ and 0.0418, respectively). Despite the downward trend, the number of atretic follicles did not decrease significantly after MLT addition (Fig. 2f, g).

**In vivo MLT intake inhibited follicle activation through PI3K-AKT-FOXO3 pathway.** To study the effect of MLT on primordial follicle activation in vivo, the mice were injected with different doses of MLT starting from PD3 to PD9 (Fig. 3a). The result showed that 1 and 15 mg kg$^{-1}$ doses of MLT significantly reduced the number of activated follicles ($P = 0.0032$ and 0.0005, respectively). It indicated that exogenous MLT intake can inhibit follicle activation. Despite the downward trend, MLT had no significant impact on follicular atresia (Fig. 3b). For the convenience of research, we used 15 mg kg$^{-1}$ as the optimal dose of MLT for subsequent experiments.

The mTORC1 and PI3K-AKT-FOXO3 are the pivotal pathways that control primordial follicle activation. In general, FOXO3 localizes to the oocyte nucleus in the dormant follicle, but exports to the cytoplasm to activate primordial follicle when phosphorylated by PI3K-AKT pathway, which can be activated

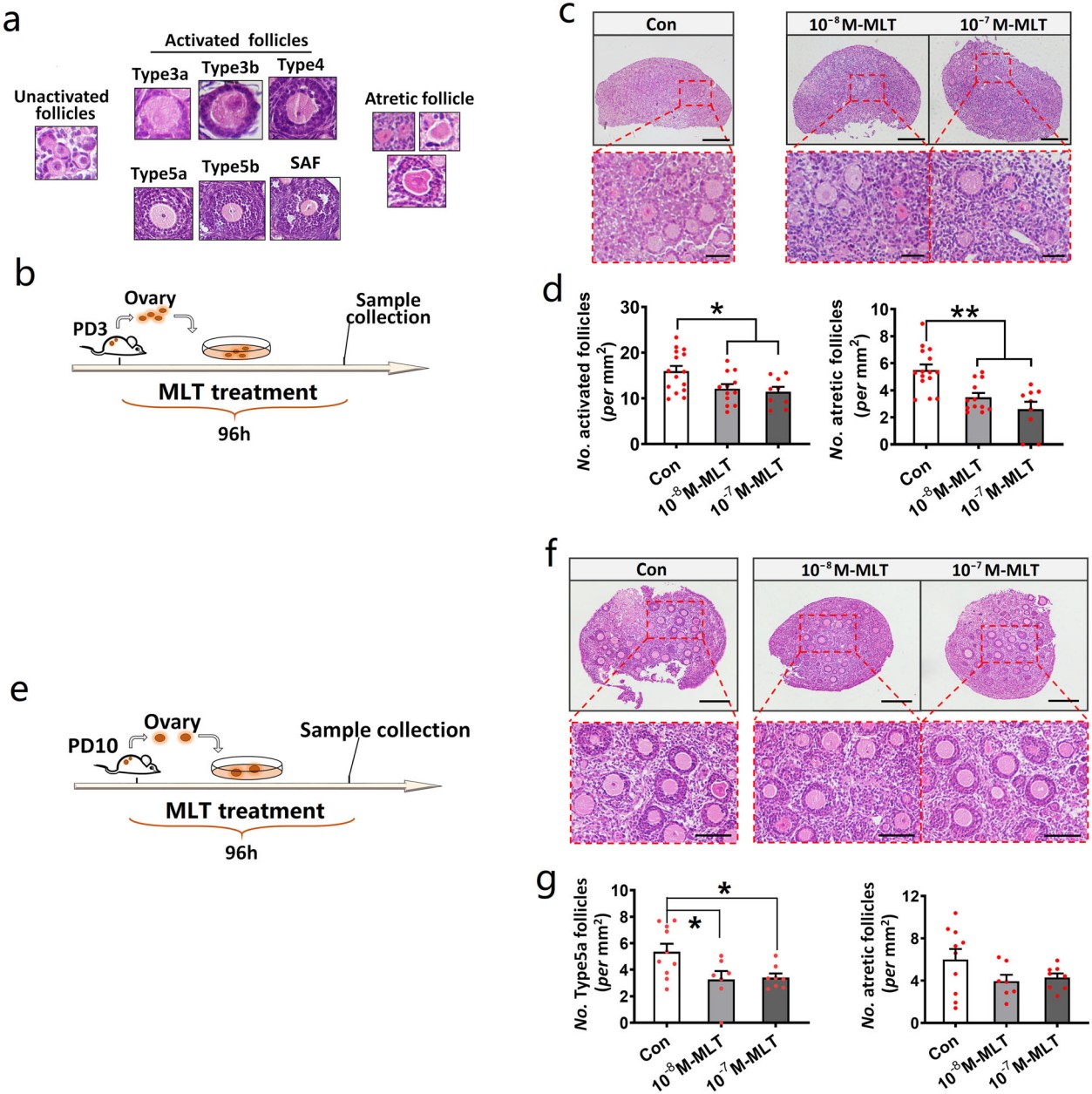

**Fig. 2 MLT addition inhibited early folliculogenesis in in vitro ovary culture system. a** Identification standards of the activated and atretic follicles. **b–d** The effect of MLT addition on follicle activation and atresia. $n = 5$ (Con), 4 ($10^{-8}$ M MLT), and 3 ($10^{-7}$ M MLT) biologically independent ovaries, respectively. **b** Experimental design. **c** Representative photographs of H&E staining in each group. The scale bars are 100 μm in the original images and 20 μm in the enlarged images. **d** The statistical charts of activated and atretic follicles (values are mean ± SEM). The number of sections used for statistics: Con $n = 15$; $10^{-8}$ M MLT $n = 12$; $10^{-7}$ M MLT $n = 9$. **e–g** The effect of MLT addition on early follicle growth and atresia. $n = 5$ (Con), 4 ($10^{-8}$ M MLT and $10^{-7}$ M MLT) biologically independent ovaries, respectively. **e** Experimental design. **f** Representative photographs of H&E staining. The scale bars are 200 μm in the original images and 80 μm in the enlarged images. **g** The statistical charts of follicles beyond type 5a-stage and atretic follicles (values are mean ± SEM). The number of sections used for statistics: Con $n = 10$, $10^{-8}$ M MLT $n = 7$, $10^{-7}$ M MLT $n = 8$. Statistical significance was determined using one-way ANOVA followed by Tukey's post hoc test. Significant differences are denoted by *$P < 0.05$ and **$P < 0.01$.

by mTORC1 pathway[36,37]. Immunofluorescent staining experiment demonstrated that MLT intake significantly inhibited nuclear exclusion of FOXO3 (PD6: $P = 0.022$; PD9: $P = 0.0235$) (Fig. 3c, d). Moreover, the data of western blotting revealed that MLT intake significantly inhibited the phosphorylation of Akt ($P = 0.0314$), although it did not affect mammalian target of rapamycin (mTOR) pathway. We further investigated whether MLT inhibits follicle activation through the Hippo pathway, which has been proved to be involved in inhibiting follicle activation and growth[38–40]. The data demonstrated that MLT

intake did not affect Hippo pathway, as the expression of key genes in Hippo pathway (*YAP*, *LATS1*, *LATS2*, *TAZ*, *MOB1b*, and *SAV1*) (Supplementary Fig. 1) and phosphorylation of Yap did not change (Fig. 3e). We also studied whether the inhibiting effect of MLT on follicle activation is mediated by transmembrane receptors. It was observed that *Mtnr1a* but not *Mtnr1b* was expressed in neonatal mouse ovary and the constitutive expression level of *Mtnr1a* was not high (Supplementary Fig. 2a). Later, we added a competitive inhibitor of Mtnr1a—Luzindole, ten times the concentration of MLT—to the in vitro culture

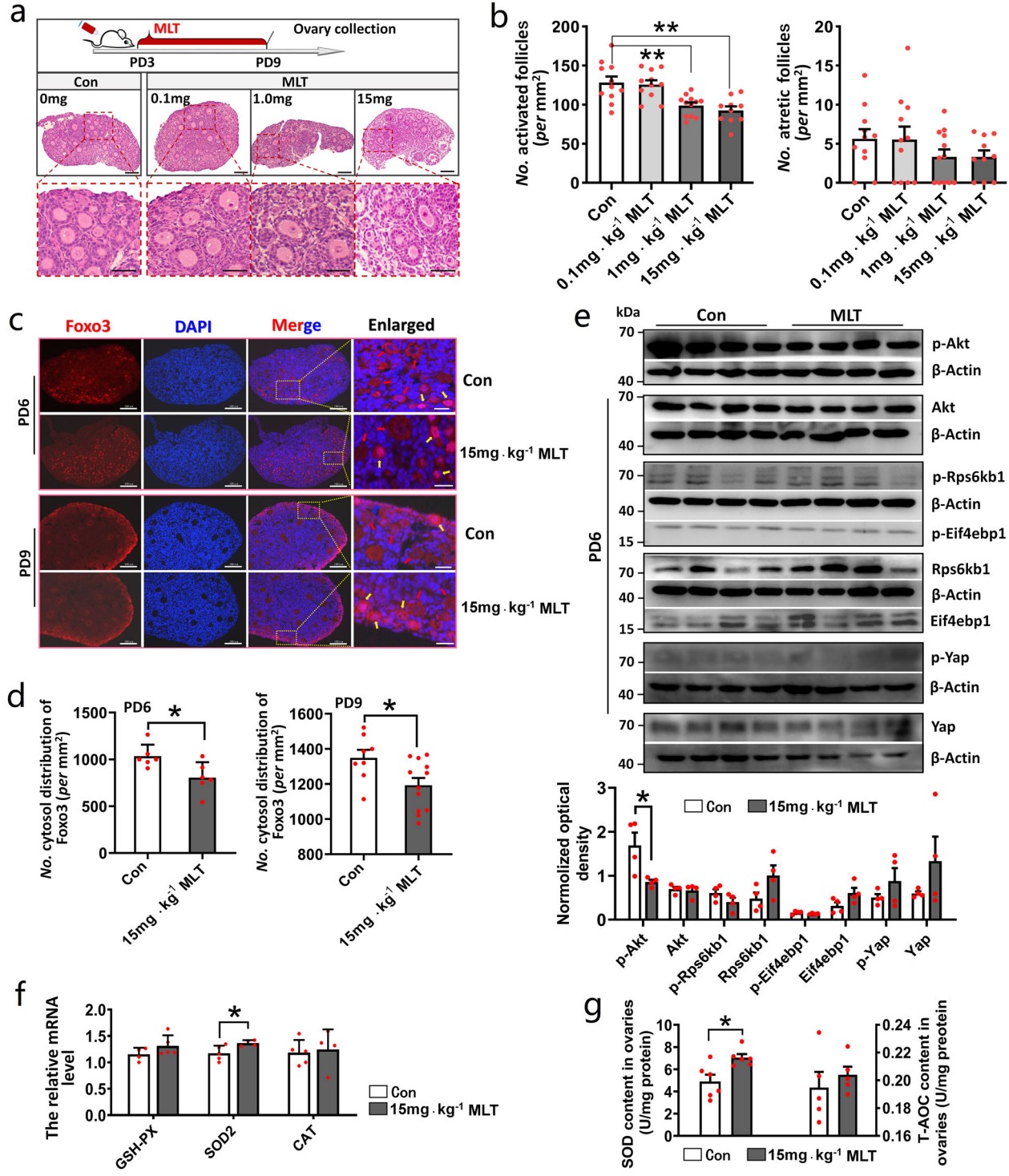

system, which still could not offset the inhibiting effect of MLT on follicle activation (Supplementary Fig. 2b, c).

We also investigated the antioxidant capacity of MLT and observed that short-term MLT intake significantly upregulated the expression of superoxide dismutase-2 (SOD2) in the ovaries ($P = 0.042$) (Fig. 3f). Similar to the gene expression data, the level of SOD significantly increased after MLT intake ($P = 0.0114$) (Fig. 3g). In addition, the serum levels of glutathione peroxidase (GSH-PX), SOD, catalase (CAT), and total antioxidant capacity (T-AOC) increased after MLT intake ($P = 0.0000$). In contrast, molondialdehyde (MDA), the metabolite of lipid peroxidation products significantly decreased after MLT intake ($P = 0.0000$)

(Supplementary Fig. 3). These data all indicated that MLT can effectively improve the antioxidant capacity of the ovary, which is consistent with previous report[27].

**In vivo MLT intake inhibited early follicle growth and atresia.** To study the effect of MLT on early follicle growth and atresia in vivo, we performed continuous MLT injection on mice from PD10 and collected ovaries at PD15, 17, 19, and 21 for histological analysis (Fig. 4a). At PD15, the number of type 5b follicles in MLT intake group was significantly reduced ($P = 0.0038$); at PD17, the number of type 5b follicles and atretic follicles in MLT intake group was significantly reduced ($P = 0.0017$ and 0.0000, respectively); the

**Fig. 3 In vivo MLT intake inhibited follicle activation through PI3K-AKT-FOXO3 pathway. a, b** The effect of MLT injection on follicle activation and atresia. $n = 3$ biologically independent mice. **a** Experimental design and representative photographs of H&E staining in each group. The scale bars are 100 μm in the original images and 50 μm in the enlarged images. **b** The statistical charts of activated and atretic follicles (values are mean ± SEM). The number of sections used for statistics: Con $n = 11$; 0.1 mg kg$^{-1}$ MLT $n = 11$; 1 mg kg$^{-1}$ MLT $n = 12$; 15 mg kg$^{-1}$ MLT $n = 10$. Statistical significance was determined using one-way ANOVA followed by Tukey's post hoc test. **c, d** The effect of MLT injection on FOXO3 translocation in the oocytes at PD6 and PD9. $n = 2$ biologically independent mice. **c** Representative photographs of immunofluorescent staining. The red arrow indicates cytosolic localization, the yellow arrow indicates nuclear localization. The scale bars are 100 μm in the original images and 20 μm in the enlarged images. **d** The statistical charts of FOXO3 distribution (values are mean ± SEM). The number of sections used for statistics: PD6 $n = 6$; PD9 $n = 8$ (Con), 12 (15 mg kg$^{-1}$ MLT). Statistical significance was determined using two-tailed unpaired Student's $t$-test. **e** The effect of MLT injection on mTOR, PI3K-AKT, and Hippo pathways in ovaries. The relative optical density was normalized by the amount of the loading control β-Actin on the same membrane (values are mean ± SEM). The original western blottings can be viewed in Supplementary Fig. 5. $n = 4$ biologically independent mice. Statistical significance was determined using two-tailed unpaired Student's $t$-test. **f** The effect of MLT injection on the expression of antioxidant genes in the ovaries (values are mean ± SEM). *GSH-PX*: $n = 4$ (Con), 5 (15 mg kg$^{-1}$ MLT); *SOD2*: $n = 5$ (Con), 4 (15 mg kg$^{-1}$ MLT); *CAT*: $n = 5$ (Con), 4 (15 mg kg$^{-1}$ MLT). Normalization was performed using the housekeeping gene *Actb*. Statistical significance was determined using two-tailed unpaired Student's $t$-test. **g** The effect of MLT injection on the levels of SOD and T-AOC in the ovaries (values are mean ± SEM). $n = 6$ (SOD), 5 (T-AOC) biologically independent samples, respectively. Statistical significance was determined using two-tailed unpaired Student's $t$-test. Significant differences are denoted by $*P < 0.05$ and $**P < 0.01$.

number of SAFs and atretic follicles was significantly reduced at PD19 ($P = 0.0064$ and 0.0000, respectively) and PD21 ($P = 0.0012$ and 0.0097, respectively) (Fig. 4b). It was also found that the diameter of oocytes in early growing follicles did not show significant differences after MLT intake (Fig. 4c). During early folliculogenesis, *Fshr* and *Lhcgr* are mainly expressed in follicles at the phase of type 5b and SAF; therefore, the expression levels of the two genes can indirectly reflect the number of follicles at the phase of type 5b and SAF in the ovary. The results of qRT-PCR showed that MLT intake reduced the expression levels of *Fshr* and *Lhcgr* in the ovaries at PD17 (*Fshr*: $P = 0.0052$; *Lhcgr*: $P = 0.09$) and PD19 (*Fshr*: $P = 0.0392$; *Lhcgr*: $P = 0.0387$) (Fig. 4d).

The formation of small follicular antrum means that they can respond to GTH stimulation, so the number of ovulated oocytes indirectly reflects the number of SAFs in the ovary. Therefore, to obtain more phenotypic evidences that MLT inhibits early follicle growth, superovulation experiment was conducted. It was observed that, at PD19, the number of ovulated oocytes in MLT intake group was significantly lower than that in control group ($P = 0.043$) (Fig. 4e, f). In addition, the decrease in the number of SAFs means the decrease in the number of follicles involved in cyclic recruitment, which indirectly leads to the decrease in the number of implanted embryos after mating. Therefore, to further demonstrate the inhibitory effect of MLT on early follicle growth, after MLT intake on PD10-21, mice were mated, then number of implanted embryos was counted (Fig. 4g). The results showed that the number of implanted embryos in MLT group was significantly smaller than that in the control group ($P = 0.0354$) and the embryo size was bigger than that in control group ($P = 0.0000$). In addition, the embryos in the MLT group showed unequal distribution on both uterine horns (Fig. 4h).

The above data indicated that exogenous MLT intake inhibits early follicle growth and atresia. To further explore the possible molecular mechanism, we performed RNA-sequencing (RNA-Seq). The volcano map of differential expressed genes was listed in Fig. 4i. Gene Ontology (GO) analysis showed that the differentially expressed genes were clustered into "Negative regulation of cell proliferation and differentiation" and "Regulation of cellular response to growth factor stimulus" (Fig. 4j). We further validated the accuracy of GO analysis. The result revealed that GC proliferation was suppressed after MLT intake, as *PCNA*, the proliferation marker, was downregulated in the MLT group ($P = 0.029$) (Fig. 4k, l). The Kyoto Encyclopedia of Genes and Genomes (KEGG) heatmap showed that the differentially expressed genes were mainly enriched in the typical pathways associated with folliculogenesis, including PI3K-AKT signaling,

mitogen-activated protein kinase signaling, vascular endothelial growth factor signaling, and transforming growth factor-β signaling pathway (Fig. 4m, n). We then used western blotting to validate KEGG analysis and the result demonstrated that the phosphorylation of Akt significantly decreased in MLT group ($P = 0.0023$) (Fig. 4o).

**SNAT-KO in mice accelerated the exhaustion of ovarian follicle reserve and age-related fertility decline.** To evaluate the effect of endogenous MLT on early folliculogenesis, we prepared *SNAT* knockout (KO) mice, among which a total of 1440 bp genomic sequences in *SNAT* was deleted (Fig. 5a, b). In serum samples from wild-type mice, MLT absorption peak appeared in the chromatography, whereas in *SNAT*-KO mice, MLT absorption peak was invisible (Fig. 5c, d). Subsequently, we evaluated the effect of *SNAT*-KO on body development, visceral indexes, and blood biochemical indexes. The results showed that body development and visceral indexes (except spleen) were not affected by *SNAT*-KO (Supplementary Fig. 4a, b). In contrast, mean corpuscular volume (MCV), spleen weights, and the number of white blood cells (WBCs), and lymphocytes (Lym) were significantly reduced ($P = 0.0062$, 0.0012, 0.0295, and 0.0072, respectively); the number of neutrophils (Neu) significantly increased ($P = 0.0114$) (Supplementary Fig. 4c). The findings may demonstrate that MLT plays an important role in immune modulation. We then evaluated the effect of *SNAT*-KO on follicle activation and atresia. It was found that the number of activated follicles significantly increased ($P = 0.039$). After MLT supplementation, the increase in activated follicles caused by *SNAT*-KO can be completely rescued ($P = 0.0000$). Similarly, the number of atretic follicles in the *SNAT*-KO ovary significantly increased ($P = 0.0064$). After MLT intake, the number of atretic follicles in *SNAT*-KO mice decreased significantly ($P = 0.0000$) (Fig. 5e, f). Consistent with the changes in atretic follicles, terminal deoxynucleotidyl transferase dUTP nick end labeling (TUNEL) staining experiment validated that apoptosis could be obviously observed in the ovary of both wild-type and *SNAT*-KO mice, but after MLT intake, apoptosis was reduced (Fig. 5g). We also examined the effect of *SNAT*-KO on the distribution of FOXO3 in oocytes. It was observed that *SNAT*-KO significantly enhanced the transport of FOXO3 to the ooplasm ($P = 0.0008$). In contrast, MLT intake effectively alleviated the effect of *SNAT*-KO on FOXO3 transport ($P = 0.0000$) (Fig. 5h, i). Furthermore, the data of western blotting also showed that MLT intake reduced the level of p-Akt in ovaries of *SNAT*-KO mice (Fig. 5j).

Then we analyzed the changes of litter size of mice with age after *SNAT*-KO. At the age of 2–3 months, *SNAT*-KO had no

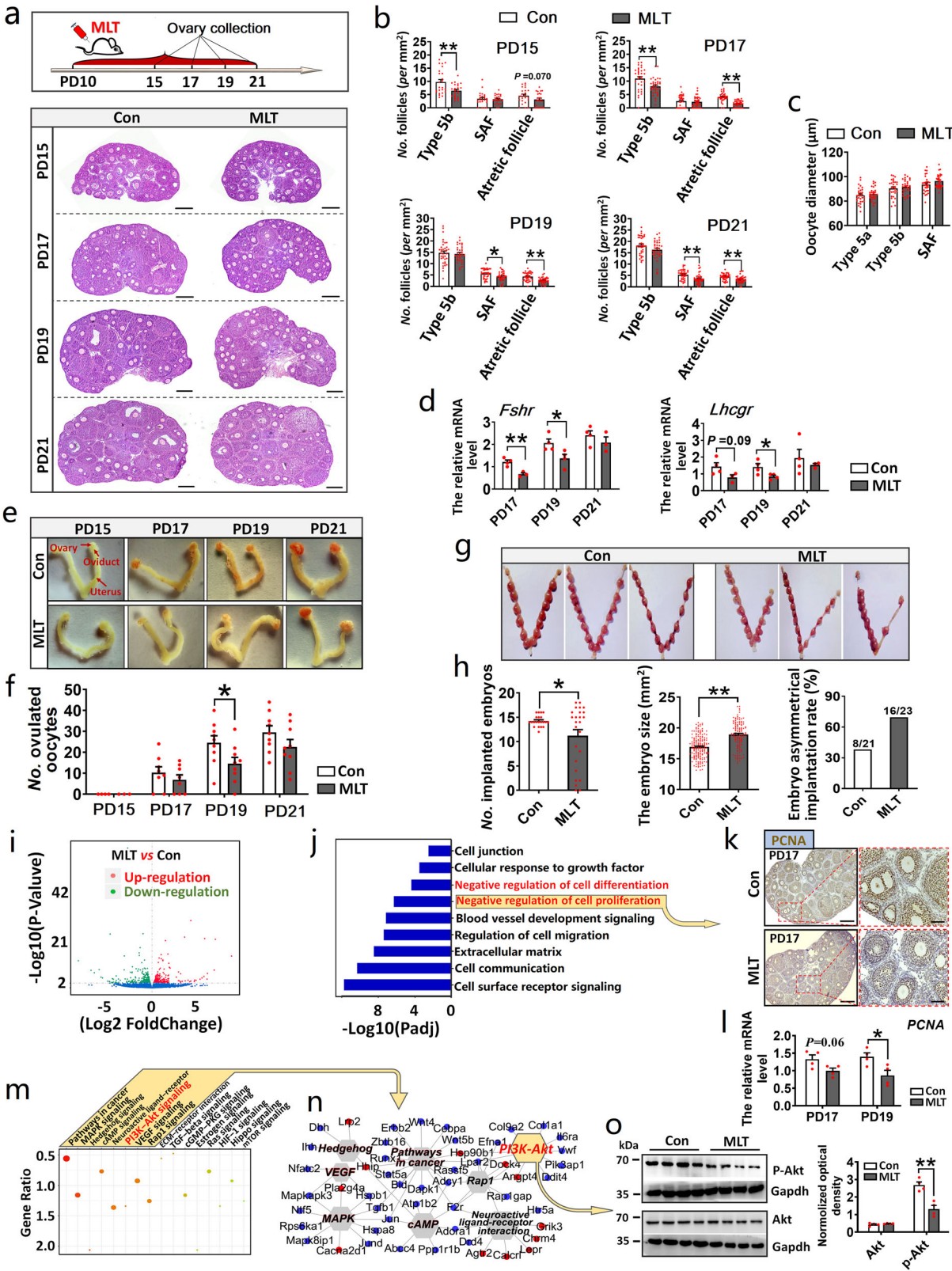

obvious effect on the litter size; at the age of 5–6 or 8–9 months, the litter size in *SNAT*-KO mice was significantly smaller than that in the wild type ($P = 0.0398$ and 0.0028, respectively) (Fig. 5k, l). The extent of ovarian aging can be reflected by the number of growing follicles in DOR, so we counted the number of growing follicles in the ovaries of 11-month-old mice. Compared with the wild type, the *SNAT*-KO mice showed a

significant decrease in the number of follicles ($P = 0.0018$) (Fig. 5m, n). *Nobox*, *Ddx4*, and *Figla*, which were solely expressed by oocytes, were the marker genes that reflected the total number of oocytes in the ovary. Among those genes, the expression levels of *Nobox* and *Figla* in *SNAT*-KO ovary was lower than that in the wild type ($P = 0.0156$ and 0.0598, respectively) (Fig. 5o). The above phenotypic, histological, and molecular data collectively

**Fig. 4 In vivo MLT intake inhibited early follicle growth and atresia. a–c** The effect of MLT injection on early follicle growth and atresia, and oocyte growth. $n = 4$ biologically independent mice. **a** Experimental design and representative photographs of H&E staining in each group. The scale bar = 200 μm. **b** The statistical charts of the number of early growing follicles and atretic follicles (values are mean ± SEM). **c** The statistical charts of the number of oocyte diameters (values are mean ± SEM). The number of sections used for statistics: PD15 $n = 24$; PD17 $n = 33$ (Con), 35 (MLT); PD19 $n = 37$ (Con), 34 (MLT); PD21 $n = 38$ (Con) and 36 (MLT). Statistical significance was determined using two-tailed unpaired Student's $t$-test. **d** The effect of MLT injection on the expression of *Fshr* and *Lhcgr* (values are mean ± SEM). Normalization was performed using the housekeeping gene *Actb*. PD17: $n = 4$ (Con), 3 (MLT); PD19: $n = 4$ (Con), 4 (MLT); PD21: $n = 4$ (Con), 3 (MLT). Statistical significance was determined using two-tailed unpaired Student's $t$-test. **e, f** The effect of MLT injection on the number of ovulated oocytes after superovulation. **e** Representative photographs of reproductive organs after superovulation. **f** The statistical chart of the number of ovulated oocytes (values are mean ± SEM). The number of mice used for superovulation: PD15 $n = 4$ (Con), 3 (MLT); PD17 $n = 8$; PD19 $n = 9$; PD21 $n = 9$. Statistical significance was determined using two-tailed unpaired Student's $t$-test. **g, h** The effect of MLT injection on the number of implanted embryos. **g** Representative photographs of implanted embryos in each group. **h** The statistical charts of the number, size, and distribution of implanted embryos. $n = 21$ (Con), 25 (MLT) biologically independent mice, respectively. Statistical significance was determined using two-tailed unpaired Student's $t$-test or $\chi^2$-test. **i** The volcano map of genes. $n = 3$ biologically independent mice. **j** GO analysis of differentially expressed genes. **k, l** The detection of MLT's effect on PCNA using immunohistochemistry and qPCR, respectively. **k** Representative photographs of immunohistochemistry. The scale bars are 200 μm in the original images and 50 μm in the enlarged images. **l** Gene quantification of *PCNA* (values are mean ± SEM). Normalization was performed using the housekeeping gene *Actb*. $n = 4$ biologically independent mice. Statistical significance was determined using two-tailed unpaired Student's $t$-test. **m, n** KEGG analysis of differentially expressed genes. **m** KEGG analysis of differentially expressed genes. The typical pathways associated with folliculogenesis are shown in yellow box. **n** Signaling pathways interaction network. The blue dots represent downregulation and red dots represent upregulation. **o** The detection of MLT's effect on PI3K-AKT pathway using western blotting. The relative optical density was normalized by the amount of the loading control Gapdh on the same membrane (values are mean ± SEM). The original western blottings can be viewed in Supplementary Fig. 6. $n = 4$ biologically independent mice. Statistical significance was determined using two-tailed unpaired Student's $t$-test. Significant differences are denoted by *$P < 0.05$ and **$P < 0.01$.

---

indicated that MLT deficiency accelerated ovarian aging by speeding up the age-related exhaustion in ovarian follicle reserve.

**Long-term intake of excessive MLT did not disturb the reproductive rhythm and growth of mice.** In the present study, both 1 and 15 mg kg$^{-1}$ MLT were effective in inhibiting follicle activation. However, it cannot be ignored that these doses are far above the physiological dose of MLT in the plasma[41]. Therefore, the safety of high-dose MLT was assessed in this section. We first studied the effect of MLT on mice physical growth. It was observed that long-term intake of 15 mg kg$^{-1}$ MLT neither affected body development nor visceral index (Fig. 6a, b). Subsequently, we evaluated the effect of MLT on the length of the estrous cycle, mating timings, pregnancy length, and labor timings. The results showed that all above indexes were not affected by long-term intake of 15 mg kg$^{-1}$ MLT (Fig. 6c–g). To evaluate the effect of MLT on the health condition of mice, the rectal temperature and blood biochemical indexes were measured. The rectal temperature did not change (Fig. 6h). The indexes that reflect the status of red blood cells (RBCs) and platelets were not affected, including those of RBC, corpuscular volume (MCV), hemoglobin (HGB), mean corpuscular HGB concentration (MCHC), and mean platelet volume (MPV). In contrast, the indexes of granulocytic amount (W-LCC) and intermediate cell amount (W-MCC), which reflect the level of immune cells, were decreased ($P = 0.058$ and $0.0136$, respectively) (Fig. 6i).

## Discussion

MLT is an extremely ancient hormone that can be traced back to the origin of life. At present, MLT has been evolved into a pleiotropic molecule[24]. In the early photosynthetic prokaryotes, MLT presumably emerged primarily as an antioxidant and free radical scavenger[42]. For plants, MLT not only acts as an antioxidant but also as a signaling molecule to modulate pathways of ethylene, abscisic, jasmonic, and salicylic acids, and is involved in stress tolerance, pathogen defense, and delay of senescence[43]. For animals, MLT is also a hormone and the hormonal property of MLT are apparent in the regulation of biological rhythm, sleep, inflammatory response, glucose metabolism, and tumor inhibition, beside the antioxidant effect[44–47]. Based on the above

characteristics, the identification of new physiological functions of MLT has long been a hot research topic in this field. Our study revealed that MLT inhibited follicle activation, growth, and atresia, which is crucial to understanding MLT's delaying the female reproductive aging. Due to the lack of germline stem cells, female ovarian reserve cannot be replenished. The exhaustion of ovarian reserve is mainly realized through follicle activation, growth, and atresia. On one hand, the dormant follicles are awakened to the early growth stage, resulting in continuous decrease of PreOR. On the other hand, DOR formed by early growing follicles is always in a dynamic balance and the follicles are either in atresia or are freed from atresia by sufficient GTH, and translated to cyclic recruitment[48,49]. Therefore, the inhibitory effect of MLT on follicle activation and growth can effectively reduce the depletion rate of ovarian reserve.

It is generally recognized that the PI3K-AKT-FOXO3 pathway controls follicle activation. Our study confirmed that MLT significantly inhibited PI3K-AKT pathway, which we believed the molecular mechanism of MLT inhibited follicle activation and growth. This study is not the first to report the inhibitory effect of MLT on PI3K-AKT pathway. For instance, a previous study found that MLT effectively inhibits the excessive activation of PI3K-AKT pathway by cisplatin, a commonly used chemotherapeutic drug, thereby protecting and stabilizing PreOR[50]; MLT reduces inflammatory response in rheumatoid arthritis by inhibiting the PI3K-AKT pathway[51]; and MLT inhibits proliferation, invasion, and tumorigenesis of human bladder cancer cell by PI3K-AKT/mTOR/MMPs pathway[52]. Of note, our study can only conclude that MLT reduces follicle activation by inhibiting PI3K-AKT pathway, but whether it is direct or indirect inhibition is not clear yet. Unlike other common hormones, MLT can be mediated by either receptors or non-receptors[53]. Here we showed that *Mtnr1a* was expressed at a low level in neonatal mouse ovaries and receptor inhibitor of Mtnr1a did not counteract the inhibitory effect of MLT on follicle activation (Supplementary Fig. 2). Therefore, the effect of MLT is at least not mediated by its transmembrane receptors. Previous studies on MLT's receptors have confirmed that in addition to transmembrane receptors, there are numerous binding sites in the cytosol and in the nucleus, such as QR2, Calmodulin, GPR50, NQO2, RZR/RORα, VDR, etc.[54–57]. Therefore, whether MLT modulates

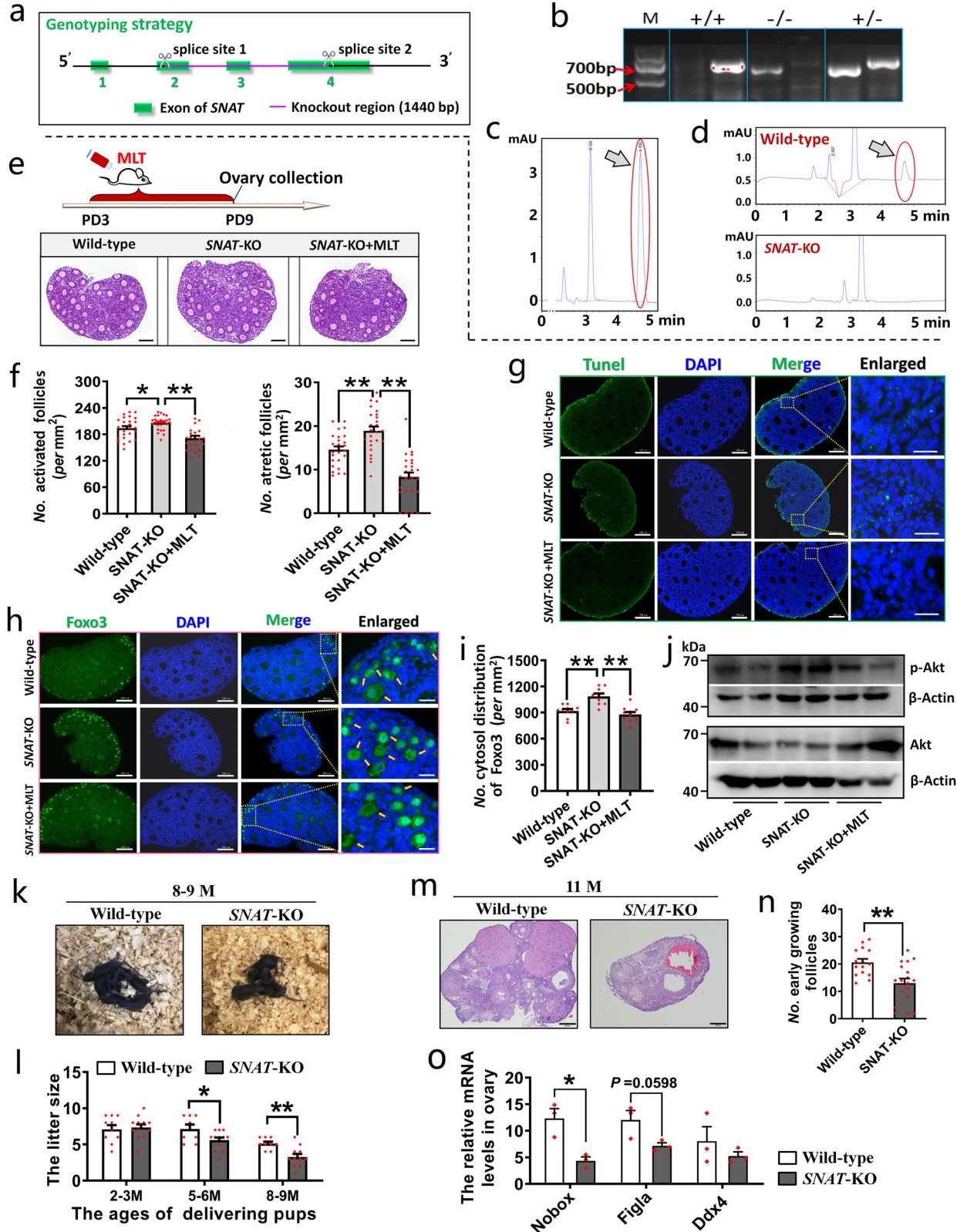

follicle activation and growth through these non-classical binding sites remains to be further investigated.

The exogenous addition (or ingestion) has been a major approach to evaluate the efficacy of MLT. For instance, MLT addition to the culture system can effectively improve the in vitro efficiency of oocyte maturation and embryo development[58]; long-term oral intake of MLT protects against cholinergic degeneration in middle-aged mice of down syndrome[59]; and long-term intake of exogenous MLT coupled with adequate sleep hygiene interventions may afford clinical benefits to children with neurodevelopmental disorders[60]. However, drug safety and public opinion requires researchers to evaluate whether endogenous MLT plays the same role to exogenous MLT. Hence, we not only investigated the effect of exogenous MLT intake on early

**Fig. 5 *SNAT*-KO in mice accelerated the exhaustion of ovarian follicle reserve and age-related fertility decline. a** Schematic diagram of *SNAT* knockout in the mouse genome. **b** The knockout mice were identified by PCR. **c**, **d** Detection of the MLT-synthetic ability of *SNAT*-KO mice. **c** The absorption peak of MLT appears in the positive control solution with MLT. **d** Representative chromatograms showed that the MLT absorption peak was visible in serum of wild-type mice but not in *SNAT*-KO mice. The positions indicated by arrows were MLT absorption peaks. **e**, **f** The effects of *SNAT*-KO and MLT intake on follicle activation and atresia. **e** Experimental design and representative photographs of H&E staining in each group, the scale bar = 100 μm. **f** The statistical chart of activated and atretic follicles (values are mean ± SEM). $n = 3$ biologically independent mice and 25 sections in each group were used for statistics. Statistical significance was determined using one-way ANOVA followed by Tukey's post hoc test. **g** Representative TUNEL staining of ovary samples in each group. The green dots represent apoptosis-positive cells. The scale bars are 100 μm in the original images and 20 μm in the enlarged images. **h**, **i** The effect of *SNAT*-KO and MLT intake on FOXO3 translocation in the oocytes. $n = 2$ biologically independent mice. **h** Representative photographs of immunofluorescent staining. The yellow arrows indicate cytosolic localization. The scale bars are 100 μm in the original images and 20 μm in the enlarged images. **i** The statistical charts of FOXO3 translocation (values are mean ± SEM). The number of sections used for statistics: wild-type $n = 10$; *SNAT*-KO $n = 9$; *SNAT*-KO + MLT $n = 10$. Statistical significance was determined using one-way ANOVA followed by Tukey's post hoc test. **j** The effect of *SNAT*-KO on PI3K-AKT pathways in ovaries. $n = 2$ biologically independent mice. The original western blottings can be viewed in Supplementary Fig. 7. **k** Representative photographs of offspring of female mice at the age of 8–9 months. **l** The effect of *SNAT*-KO on litter size (values are mean ± SEM). The number of mice used for mating: 2–3 M $n = 10$ (wild-type), 12 (*SNAT*-KO); 5–6 M $n = 8$ (wild-type), 12 (*SNAT*-KO); 8–9 M $n = 9$ (wild-type), 11 (*SNAT*-KO). Statistical significance was determined using two-tailed unpaired Student's *t*-test. **m**, **n** Morphological features of ovaries from 11-month-old *SNAT*-KO and wild-type group were analyzed. $n = 4$ biologically independent mice. **m** Representative photographs of H&E staining in each group. The scale bar = 200 μm. **n** The statistical chart of early growing follicle (values are mean ± SEM), the number of sections used for statistics: wild-type $n = 15$; *SNAT*-KO $n = 18$. Statistical significance was determined using two-tailed unpaired Student's *t*-test. **o** The expression of oocyte marker genes in *SNAT*-KO and wild-type groups (values are mean ± SEM). Normalization was performed using the proportion of reverse-transcribed RNA to total RNA. $n = 3$ biologically independent mice. Statistical significance was determined using two-tailed unpaired Student's *t*-test. Significant differences are denoted by *$P < 0.05$ and **$P < 0.01$.

folliculogenesis, but the role of endogenous MLT. Cell culture indicated that GCs had the ability to secrete MLT (Fig. 1c), the loss of follicle reserve in MLT-deficient mice was faster than that of wild type, and phenotype of ovarian aging in middle age was more obvious than wild type (Fig. 5). The above results indicated that endogenous MLT was indeed involved in the regulation of early folliculogenesis and ovarian aging. Of note, we found that MLT deficiency did not significantly affect the fecundity of young mice. We speculate that, because MLT has many physiological functions, each of which cannot completely interrupt the related physiological process. MLT is more like a lubricant for life activities: with it, the activities run more smoothly; without it, the activities can still happen and adverse impact may take a long time to appear. It should also be noted that the KO mice in this study are of C57BL/6 strain, which has a point mutation in MLT synthesis gene, resulting in a comparatively lower MLT level than other strains[61]. We selected the KO mice of C57BL/6 strain, because in this strain, low MLT level was mainly found in the pineal gland but MLT levels in peripheral tissues were little different from other strains, such as the blood, thymus, and spleen[62]. Further, our final experimental results also confirmed that *SNAT*-KO in C57BL/6 mice accelerated the exhaustion of ovarian follicle reserve and affected immunological indexes.

The pineal gland has long been considered as the main organ for MLT synthesis. However, in recent years, some peripheral organs, such as the gastrointestinal tract, liver, and spleen, have also been reported to have the ability to synthesize MLT. Moreover, MLT synthesized by peripheral organs may have local functions different from that of the pineal origin[63]. Therefore, the identification of new synthesis sites is crucial for a better understanding of MLT's function. Previous studies have found that preovulatory follicular fluid has high levels of MLT[64], oocytes synthesize MLT through mitochondria to resist free radical[65], and *SNAT* and *MT2* are expressed in abundance in human corpus luteum[66]. Our study further demonstrated that GCs can synthesize MLT (Fig. 1c). Therefore, we can reasonably speculate that ovary is another MLT-secreting organ based on the above findings. Our study also demonstrated that MLT content in neonatal ovaries decreased with day age (Fig. 1e). Interestingly, however, MLT levels in the ovaries do not always decline. It has been reported that after the follicles enter GTH-dependent phase, GTH

can upregulate MLT content in follicles before ovulation, and the increase of MLT content is involved in the modulation of luteinization[67]. Therefore, we speculate that there may exist a mutual regulation mode mediated by MLT signaling between GTH-dependent follicles and GTH-independent follicles (i.e., primordial and early growing follicle) in the ovary. On one hand, primordial and early growing follicles provide sufficient supply of follicles for GTH-dependent folliculogenesis through activation and growth. On the other hand, GTH-dependent follicles secrete MLT so as to not only regulate its own development, but also inhibit follicle activation and growth in a negative feedback manner, helping to avoid overactivation of ovarian reserve.

With the further study of follicle activation, some substances (including rapamycin, bpV, ZCL278, Metformin, and 2-DG) have been listed as candidate drugs to delay ovarian aging or treat premature ovarian failure[68–72]. For instance, rapamycin is highly recommended to be used in cancer chemotherapy for its strong inhibition of mTOR pathway. Recently, researchers have significantly prolonged the reproductive age of mice by short-term injection of rapamycin[69]; as PTEN-PI3K-AKT pathway controls follicle activation, doctors have activated the follicles in vitro by adding the PTEN (phosphatase and tensin homolog) inhibitor bpV, which successfully allowed women who have premature ovarian failure gave birth to healthy children[70,71]. However, the above methods also have limitations. During the phase of rapamycin treatment, experimental animals have estrous cycle disorder, sudden weight loss, etc.[69]; the activity of the PTEN-PI3K-AKT pathway was enhanced in dividing tumor cells and stem cells, so either artificial inhibition or activation of those pathways will bring about unpredictable physiological risks. In contrast, MLT has an advantage in terms of safety and commercial MLT health products have already been available in the market. Further, as an endogenous molecule, MLT levels in follicular fluid of human are known to decrease with age[73]. Our study demonstrated that 1-month MLT intake of high-dose had no significant effect on the estrous cycle, reproductive rhythm, and physical development of experimental animals (Fig. 6). Nevertheless, whether MLT has a potential to act as a human ovarian health product needs further investigation. In this study, the minimum dose of MLT to inhibit follicle activation is 1 mg kg$^{-1}$. If converted to a human dose, it is about 50 mg per day, but the current

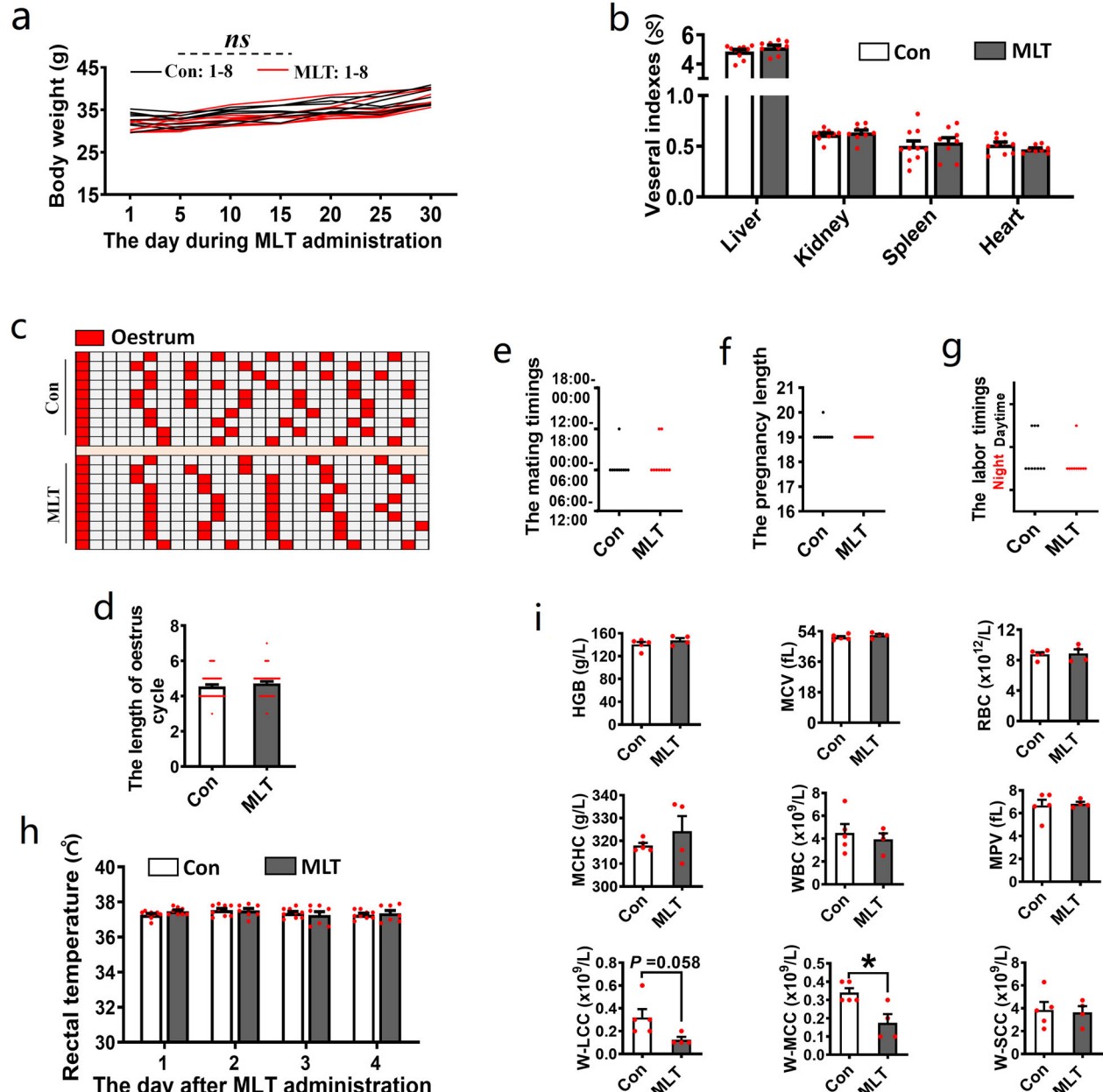

**Fig. 6 Long-term intake of excessive MLT did not disturb the reproductive rhythm, growth, and health of mice. a** The effect of MLT intake on daily weight gain. *n* = 8 biologically independent mice. **b** The effect of MLT intake on visceral indexes. *n* = 10 (Con), 9 (MLT) biologically independent mice, respectively. **c**, **d** The effect of MLT intake on estrous cycle. **c** The chart of estrus records, red grids represent estrum, and the distance between two adjacent red grids represents the length of an estrus cycle. **d** The statistical chart of estrus cycle length. *n* = 10 biologically independent mice. **e** The effect of MLT intake on mating timings of female mice. *n* = 10 biologically independent mice. **f** The effect of MLT intake on pregnancy length. *n* = 10 biologically independent mice. **g** The effect of MLT intake on labor timings. *n* = 10 biologically independent mice. **h** The effect of MLT intake on body temperature. *n* = 9 (Con), 8 (MLT) biologically independent mice, respectively. **i** The effect of MLT on plasma biochemical indexes. *n* = 5 (Con), 4 (MLT) biologically independent mice. Statistical significance was determined using two-tailed unpaired Student's *t*-test. Values are mean ± SEM. Significant differences are denoted by *$P < 0.05$. "ns" stands for $P > 0.05$, not significant.

recommended maximum intake of commercial MLT products is 10 mg per day. Therefore, the dose of MLT intake of mice cannot be applied directly to humans. In addition, both MLT deficiency and long-term MLT intake had significant effects on the immunological indexes of experimental animals. For instance, the spleen weight of MLT-deficient mice was significantly lower than that of wild-type mice and the total amount of WBCs and the proportion of Lym in the blood were significantly reduced (Supplementary Fig. 4). Therefore, before using MLT as a health supplement to delay ovarian aging, it is necessary to evaluate the

changes of human immune system function and the impact of such changes on health after long-term MLT intake.

In summary, this study demonstrates that endogenous MLT is involved in the regulation of early folliculogenesis and presents more insight into its roles in ovarian aging. In addition to its well-known antioxidant property, MLT as an endogenous molecule can also slow down ovarian aging by directly inhibiting follicle activation, growth, and atresia (Fig. 7). Furthermore, this study also provides direct evidence to confirm that the ovary is an organ that can synthesize MLT.

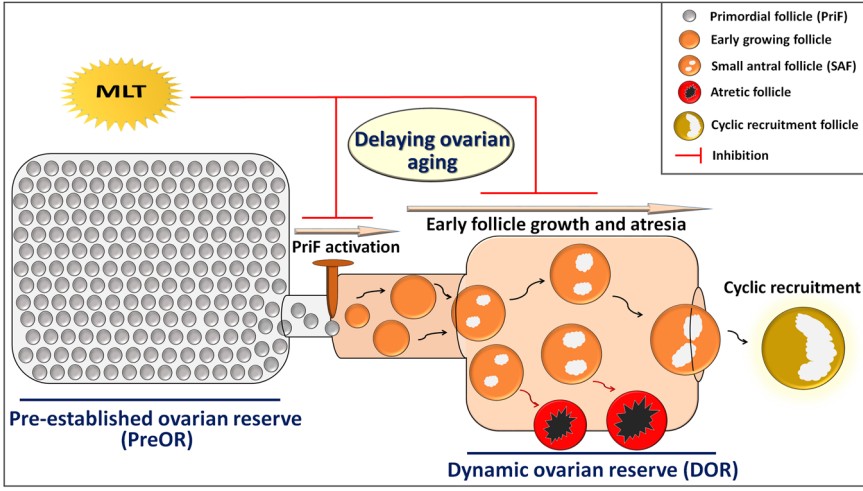

**Fig. 7 A proposed model to explain how MLT delays ovarian aging.** The depletion of ovarian follicle reserve is the direct cause of ovarian aging. MLT slows down the exhaustion of follicle reserve by inhibiting the activation, growth, and atresia of follicles in PreOR and DOR, thus delaying ovarian aging.

## Methods

**Animals.** KM-strain mice (0–4 weeks old) were purchased from the Center for Animal Testing of Huazhong Agricultural University. *SNAT*-KO mice of C57BL/6 strain (0–11 months old) were obtained from Cyagen Biosciences (Guangzhou, China). The genotyping method of identifying *SNAT*-KO mice is as follows: Primer-1 was used for first PCR screening and the amplified fragment length was 627 bp; Primer-2 was used for PCR reconfirmation and the amplified fragment length was 669 bp. The reaction procedure was as follows: initial denaturation at 95 °C for 5 min, 40 cycles at 95 °C for 30 s, annealing at 60 °C for 30 s, then extension at 72 °C for 40 s, and a final elongation step at 72 °C for 5 min. The heterozygote mice have two amplicons, whereas the homozygote mice have only one amplicon. Primer sequences are listed in Supplementary Table 1. Mice were housed in a temperature-controlled facility (24 ± 2 °C) with constant 12 h light–dark cycles and were allowed access to food and water ad libitum. All experiments and animal handling were conducted in accordance with the institutional guidelines for animal experimentation after obtaining prior approval from the Institutional Animal Ethics Committee of Huazhong Agricultural University (HZAUMO-2017-049).

**Experimental design and animal treatment.** To assess the effect of MLT on follicle activation in vitro, the ovaries from mice at PD3 were cultured with MLT (Sigma, St. Louis, MO, USA) and then sectioned for histological analysis. To assess the effect of MLT on early follicle growth in vitro, the ovaries from mice at PD10 were cultured with MLT and then sectioned for histological analysis. To assess the effect of MLT on follicle activation in vivo, MLT were injected into mice from PD3. The ovaries at PD9 were sectioned for follicle counting and homogenized for the antioxidant assay. In addition, the ovaries at PD6 and PD9 were sectioned for immunofluorescence staining and ovaries at PD6 were homogenized for western blotting. To assess the effect of MLT on early follicle growth and atresia, MLT was injected into mice from PD10. The ovaries at PD15, 17, 19, and 21 were sectioned for histological analysis and ovaries at PD17 were collected for RNA-Seq and western blot analysis. To explore the effect of *SNAT*-KO on mice fecundity, the activated and atretic follicles of juvenile mice were evaluated. Adult mice were mated with wild-type males at the ages of 2–3 months, 5–6 months, and 8–9 months, respectively, and then the litter sizes were recorded. The ovaries from 11-month-old mice were collected for hematoxylin and eosin (H&E) staining and qRT-PCR assay. To investigate the effects of *SNAT*-KO on growth and health, neonatal mice were weighed from week 1 to 8. By adulthood, the weights of organs (heart, liver, spleen, and kidney) were measured and then divided by body weight to calculate visceral index. In addition, the blood samples were collected for biochemical index assay. To evaluate the effect of MLT on growth and health, 15 mg kg$^{-1}$ MLT was injected to mice, which lasted 30 days. The rectal temperature was recorded 4 days, 4 times; the body weight was recorded every day; on the last day, the weights of organs (heart, liver, spleen, and kidney) were measured to calculate visceral index and blood samples were collected for biochemical index assay. To examine the effect of MLT on reproductive rhythm, 15 mg kg$^{-1}$ MLT was injected every day and then estrum was recorded through vaginal smear, and mating timings were also recorded. To examine the effect of MLT on pregnant rhythm, 15 mg kg$^{-1}$ MLT was injected to mice when gestational age was above 12 days and then pregnancy lengths and labor timings were recorded.

**Histomorphological analysis.** Paraffin-embedded ovaries were sectioned for H&E staining (5 μm-thick paraffin sections). Thereafter, morphometric measurements were performed using digitalized images obtained directly from the microscope

(Olympus BX53, Tokyo, Japan). Then morphometric parameters were measured using ImageJ software (National Institutes of Health, Bethesda, MD, USA). Early growing follicles at primary (type 3a, b), secondary (type 4, 5a, 5b), SAF stages of development, and atretic follicles were counted in all sections based on the well-accepted standards established by the Pedersen and Peters[74] criterion. Briefly, type 3a, b follicles were characterized by a monolayer of cubic follicle cells surrounding an oocyte; type 4 follicles have two layers of follicle cells surrounding an oocyte; type 5a follicles have three layers of follicle cells; and type 5b follicles were characterized by many layers of follicle cells surrounding a fully grown oocyte, but the follicular antrum has not yet formed[74].

**Ovary culture.** Ovaries were collected on the designated time by microdissection in preheating phosphate-buffered saline. The ovaries from mice at PD3 were cultured on an insert (PICM0RG50, Millipore, USA) in six-well culture dishes in 1500 μL Dulbecco's modified Eagle's medium/Ham's F12 nutrient mixture (DMEM/F12) (GIBCO, Life Technologies, USA) containing 0.1% bovine serum albumin (BSA) (Sigma, St. Louis, MO, USA), ITS (1 : 100, Sigma, St. Louis, MO, USA), and Penicillin–Streptomycin Solution at an incubator (5% CO$_2$, 37 °C, and saturated humidity). The ovaries from mice at PD10 were cultured in the above medium with FSH (30 ng/ml, Sigma, St. Louis, MO, USA). Ovaries were cultured in either medium alone or medium with MLT or Luzindole (Sigma, St. Louis, MO, USA).

**HPLC assay on MLT.** A 1 ml syringe was used to puncture the follicles on the ovary surface to release GCs. Isolated GCs were incubated with $10^{-5}$ M 5-HT (Sigma, St. Louis, MO, USA) or culture medium alone (DMEM/F12 containing 0.1% BSA, 1.0% ITS) for 6 h to test their MLT-biosynthetic capacity. To verify whether *SNAT*-KO leads to loss of MLT in mice, the serum was collected for MLT assay. Ovaries were collected from mice at PD7, 9, 15, 17, and 19, respectively, and then homogenized immediately in 1 mL pre-cooling methylalcohol (Sigma, St. Louis, MO, USA) and centrifuged at 12,000 r.p.m. for 10 min at 4 °C. The supernatants were collected for MLT assay. The fragments of ovarian tissues were lysed with RIPA lysis buffer and total proteins were measured with BCA Protein Assay Kit (CWBiotech, Beijing, China) to normalize MLT level.

Chromatographic MLT separation was performed on Agilent 1260 Infinity II HPLC system (Agilent, America) equipped with a variable wavelength detector. The assay was carried out on InertSustain C18 column (250 mm × 4.6 mm, 5 μm, GL Sciences, Inc., Japan) with acetonitrile/ultrapure water (4/6) mixture flowing at a rate of 1 mL/min. Sample temperature, injection volume, and detection wavelength was 25 °C, 20 μL, and 222 nm, respectively. The MLT standard was purchased from Sigma, and was subsequently diluted to different concentrations for testing the standard curve. MLT in the samples was identified by its peak retention time compared to those of standards and were quantified based on peak height.

**Measurements of antioxidant activity and lipid peroxidation.** Ovaries of each group were collected and homogenized. The levels of SOD and T-AOC were measured by colorimetry. Blood samples of each group were collected and after clotting for 30 min, the serum was obtained by centrifugation at 3000 r.p.m. for 10 min and stored at −20 °C. The levels of GSH-PX, CAT, SOD, MDA, and T-AOC were measured by colorimetry. The detection was entrusted to Beijing North Institute of Biological Technology (Beijing, China). The detection kits were all purchased from Jiancheng Bioengineering Institute (Nanjing, China).

**Superovulation, mating, and embryo count**. Mice were injected with 5 I.U. PMSG (Ningbo Hormone Products Co., Ltd, Zhejiang, China) to stimulate follicle growth. A dose of 5 I.U. hCG (Ningbo Hormone Products Co., Ltd, Zhejiang, China) was then injected to trigger ovulation after 44 h. Subsequently, mice were killed, the reproductive organs were collected for visual observation, and ovulated oocytes were collected by puncturing the oviduct for quantitative analysis. Female mice were caged with adult males. The day of mating was recorded as the first day of gestation. Female mice were killed at 8.5 gestation day. Then the uteri were collected for photographic documentation, and the sizes and distribution of implanted embryos were measured.

**Measurement of biochemical indexes**. The whole blood samples in each group were collected in anti-coagulation tubes. Thereafter, the biochemical indexes were analyzed by a blood cell analyzer (pocH-100iV Diff, Sysmex, Japan). The detection was entrusted to the Veterinary Hospital of Huazhong Agricultural University. The test indexes included RBC count, corpuscular volume (MCV), HGB concentration, MCHC, MPV, WBC count, Lym, Neu, Lym count (W-SCC), granulocytic amount (W-LCC), and intermediate cell amount (W-MCC).

**RNA-Seq and data analysis**. The sequencing and data annotation were completed by Novogene Co., Ltd (Beijing China). Briefly, total RNAs were extracted using the Trizol reagent. Sequencing libraries were generated using NEBNext® UltraTM RNA Library Prep Kit for Illumina® (NEB, USA). To select cDNA fragments of preferentially 250–300 bp in length, the library fragments were purified with the AMPure XP system (Beckman Coulter, Beverly, USA). Then, the library quality was assessed on the Agilent Bioanalyzer 2100 system. Gene expression was normalized as the fragments per kilobase of exon per million fragments mapped (FPKM). FPKM values were calculated in each RefSeq gene for differential expression analysis. The $P < 0.01$ obtained with the Audic Claverie test was considered to be statistically significant. GO was used to annotate biological terms and the KEGG was used to find the associated pathways.

**Immunohistochemistry and immunofluorescence**. For immunohistochemistry, the sections were incubated with rabbit polyclonal antibodies (SNAT: 1 : 100; ab3505, Abcam; PCNA: 1 : 100, GT2253, Genetech, Shanghai, China) overnight at 4 °C; then, the sections were incubated with secondary antibody (goat anti-rabbit IgG: 1 : 200, G1213; goat anti-mouse IgG: 1 : 200, G1214, Servicebio Technology, Wuhan, China) for 1 h at 37 °C. After washing, sections were incubated with streptavidin–horseradish peroxidase (CWBiotech, Inc., Beijing, China) for 2 h. Target protein was visualized using DAB (Sigma-Aldrich, Shanghai, China) as the chromogen. For immunofluorescence staining, the sections were incubated with rabbit monoclonal antibodies (FOXO3a: 1 : 200, 2497 S, CST) overnight at 4 °C. Afterwards, the sections were incubated with fluorescent secondary antibody (goat anti-rabbit IgG: 1 : 200, GB21303, Servicebio Technology, Wuhan, China) for 50 min and subsequently stained with 4′,6-diamidino-2-phenylindole (DAPI) for 5 min. The images were obtained directly from ortho-fluorescence microscopy (Olympus, Co., Japan).

**Western blotting**. Total proteins were extracted with RIPA lysis buffer (Servicebio Technology, Wuhan, China). Thereafter, proteins were separated via SDS–polyacrylamide gel electrophoresis and then were subsequently transferred to polyvinylidene fluoride membranes. Subsequently, the membranes were incubated with primary antibodies for the following proteins: Akt (1 : 1000; 9272S, CST), p-Akt (Ser473; 1 : 1000; 4060S, CST), Rps6kb1 (1 : 1000; 9202S, CST), p-Rps6kb1 (Thr-389; 1 : 1000; 9205S, CST), Eif4ebp1 (1 : 1000; 9452S, CST), p-Eif4ebp1 (Ser-65; 1 : 1000; 9451S, CST), Yap (1 : 1000; 14074S, CST), p-Yap (Ser-127; 1 : 1000; 13008S, CST), Gapdh (1 : 2000; CW0100, CWBiotech, Beijing, China), and β-Actin (1 : 2000; CW0096, CWBiotech, Beijing, China). The membranes were then incubated with the corresponding secondary antibody (goat anti-rabbit IgG: 1 : 4000; BF03008, Biodragon-immunotech, Beijing, China; goat anti-mouse IgG: 1 : 4000, BF03001, Biodragon-immunotech, Beijing, China) for 2 h. Finally, the immunoblots were visualized with an ECL kit (CWBiotech, Beijing, China). The optical density of phosphorylated proteins (p-AKT, p-Rps6kb1, p-Eif4ebp1, and p-Yap) was measured by ImageJ software (National Institutes of Health, Bethesda, MD, USA) and normalized by the amount of the loading control on the same membrane.

**PCR analysis**. Total RNA was extracted using the Trizol reagent (Invitrogen, Inc., Carlsbad, CA, USA). Reverse transcription was carried out by using the PrimeScript™ RT reagent kit with genome DNA Eraser (Takara Bio, Inc., Dalian, China). RT-PCR were run using a 2× Taq Plus PCR MasterMix (Tiangen Biotech, lnc., Beijing, China.). The reaction procedure was as follows: initial denaturation at 95 °C for 5 min, 40 cycles at 95 °C for 30 s, annealing at 64 °C for 30 s, then extension at 72 °C for 40 s, and a final elongation step at 72 °C for 5 min. Real-time quantitative PCR (qPCR) were run using a QuantiFast SYBR Green PCR Kit (Takara Bio, Inc., Dalian, China). The qPCR procedure was as follows: an initial step of 2 min at 50 °C, denaturation at 95 °C for 3 min; 35 cycles at 95 °C for 5 s, and 60 °C for 20 s, and a final elongation step at 72 °C for 3 min. Normalization was performed using the housekeeping gene *Actb*. The relative mRNA level was calculated by the $2^{-\triangle\triangle\text{ct}}$ method. When *Nobox*, *Figla*, and *Ddx4* were used to evaluate the oocyte count in the ovary, the normalization was performed using the proportion of reverse-transcribed RNA to total RNA. Primer sequences are listed in Supplementary Table 1.

**TUNEL staining**. TUNEL staining was utilized to detect apoptosis in the ovaries from wild-type, *SNAT*-KO, and *SNAT*-KO + MLT groups according to the manufacturer's protocol (In Situ Cell Death Detection Kit, Roche, Germany). Briefly, paraffin-embedded ovaries were subjected to routine 5 µm thickness sectioning. All sections were incubated in TUNEL reaction medium for 2 h at 37 °C in the dark. After the reaction was stopped, the sections were washed and stained with DAPI for 5 min. The TUNEL-positive cells in ovarian tissues from all the groups were observed using ortho-fluorescence microscopy (Olympus, Co., Japan).

**Statistics and reproducibility**. Statistical analyses were performed using GraphPad Prism 7 (GraphPad, San Diego, CA, USA). One-way analysis of variance followed by Tukey's post hoc test and two-tailed unpaired Student's *t*-test was used to analyze the statistical significance among multiple groups and between two groups, respectively. The comparison between the percentages used the $\chi^2$-test. *P*-value < 0.05 was considered statistically significant. The reproducibility was determined by using several biological replicates as indicated in the figure legends. Conclusions were not determined by a single detection technique, but by the combination of several experimental techniques from different perspectives. When performing histological analysis, we assigned three individuals who had not known grouping information to independently record and obtained similar results.

**Reporting summary**. Further information on research design is available in the Nature Research Reporting Summary linked to this article.

## Data availability

The RNA-sequencing data are openly available in Zenodo (https://doi.org/10.5281/zenodo.3365397)[75]. Source data underlying plots shown in figures are provided in Supplementary Data 1. All other data are available upon reasonable request.

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

## Acknowledgements

This research was supported by the National Natural Science Foundation of China (31701301), the National Key Research and Development Program of China (2017YFD0501701), and the Fundamental Research Funds for the Central Universities of China (2662018PY037).

## Author contributions

C.H. conceived the project, analyzed the data, and wrote the manuscript. C.Y. and Q.L. helped to design the study, performed the experiments, participated in analyzing the dataset, and writing the manuscript. Y.C., X.W., Z.R., and F.F. assisted with sample collection and data analysis, and revised the manuscript. J.X., G.L., X.L., and L.Y. reviewed the manuscript and provided substantial advice through experimental work. All authors read though the manuscript and approved the final version.

## Competing interests

The authors declare no competing interests.
