## [Peer Review File · Communications Biology]

Reviewers' comments:

Reviewer #1 (Remarks to the Author):

This paper provides some evidence that melatonin, either endogenously or exogenously can affect the rate of activation of primordial follicles in the mouse ovary, with potential implications for reproductive lifespan. The authors provide evidence for melatonin synthesis in the granulosa cells of the ovary, pharmacological affects on follicle activation and importantly evidence from a SNAT knockout mouse.
Specific Comments

1. There are a substantial number of areas of English usage throughout the paper that should be addressed. There are also a number of rather surprising statements, starting with the very first sentence suggesting that ovarian aging is a gain of function mutation. This seems a very strange comment that should be reconsidered. Similarly, on line 34, there is evidence that lots of gonadal hormones triggers cancer but the references cited provide no evidence for this.
2. Line 35. A further similar strange comment is describing ovarian aging as a pacemaker for female body aging.
3. Line 81. VC and VE presumably referred to the vitamins.
4. Figure 1 shows SNAT localization in the ovary. The day 9 and day 17 images show very poor ovarian architecture and should be replaced with better quality fixed tissue. This is also evidenced in the loss of oocytes from the sections that are shown. A wide range of follicles could also be shown including antral follicles. Negative controls are also not described.
5. There is no analysis of the presence of melatonin receptors within the ovary, which is fundamental to the hypothesis presented.
6. The decrease in SNAT mRNA may merely reflect dilution caused by the presence of larger antral follicles and these data would be much more informative if they were expressed relative to another marker of the small follicle population that melatonin is proposed to regulate. This was applied to panel D in figure 1.
7. The authors use the term 'extremely significant' repeatedly through the manuscript, this should be changed to just 'significant'.
8. In figure 2, data on the number of primordial follicles should also be presented as it is possible that the survival of that population might be affected. This is very relevant to interpretation of the number of activated follicles.
9. Line 209. The authors should provide evidence of lack of activity of the PI3K pathway and changes to localization of foxo3. This does not demonstrate that these pathways are the target of melatonin action on primordial follicles as changes in their activity is a reflection of the changes in follicular activity rather than necessarily showing a direct targeting.
10. Line 218. Changes in various serum anti-oxidant markers are demonstrated, without changes in the mRNA activity of the relevant genes in the ovary. The authors should be more cautious as to interpreting this to mean that melatonin improves the anti-oxidant capacity of the ovary.
11. Figure 4. The histological data are calculated as proportion of different follicle types relative to number of activated follicles. I am not at all convinced by the validity of this analysis and data on the absolute numbers of the different follicle types would be a lot more robust. Similarly the interpretation of the data on the expression levels of FSH and LH receptor is unconvincing, and is difficult to interpret reliably.
12. Figure 4, panel E. The heatmap is inadequately labelled and is essentially not interpretable.
13. Line 371. In addition to follicle numbers, ovarian weights would also be informative here.
14. Line 372. GDF9 and ZP3 are not expressed by oocytes and primordial follicles but rather by oocytes within growing follicles.
15. Line 440. There is now some discussion of cisplatin out of the blue in this manuscript, which is very confusing as it has not been mentioned earlier in the manuscript. The authors should carefully consider the organisation of the discussion.
16. Line 474. I do not think the authors have investigated premature ovarian failure in this study and this comment should be removed.
17. Line 484. Why did the authors choose this specific mouse strain which already has a mutation in

melatonin synthesis?

18. Figure 7 has a spelling mistake in 'delaying'.

Reviewer #2 (Remarks to the Author):

Review of the manuscript COMMSBIO-20-2920, entitled " Melatonin as an endogenous hormone to slow down the exhaustion of ovarian follicle reserve in mice—a novel insight into its roles in early folliculogenesis and ovarian aging."

Overview and general comments

The authors investigated the role of melatonin on ovarian aging and the mechanism of melatonin to delay ovarian aging. In vitro and in vivo experiments showed that melatonin inhibited primordial follicle activation by inhibiting the PI3K-AKT-FOXO3 pathway, and melatonin also inhibits early follicle growth and atresia. They also confirmed that SNAT knockout in mice resulted in a significant increase in follicle activation and atresia, and eventually accelerated ovarian aging. They concluded that endogenous melatonin is involved in the regulation of ovarian aging, and reveals that melatonin delays ovarian aging by inhibiting primordial follicle activation, early follicle growth and atresia. This study is variable to understand the role of melatonin on ovarian physiology and ovarian aging. They demonstrated that not only direct anti-oxidative effect of melatonin but also novel mechanism of melatonin to inhibit primordial follicle activation are involved in the regulation of ovarian aging. The study design is well refined.

Major comments

1. Melatonin produced by pineal gland is secreted into the blood circulation, however, extra-pineal melatonin produced by other organs stay inside the cell and protects themselves from oxidative stress. If primordial and early follicles produce melatonin, how melatonin inhibits primordial activation? Autocrine action? Paracrine action? Does melatonin inhibit primordial follicle activation themselves or other primordial follicles? How many animals per group in each experiment?

2. Melatonin inhibits primordial follicle activation by suppressing PI3K-AKT-FOXO3 pathway. It may through melatonin membrane receptor (MT1, MT2) or melatonin nuclear receptor (ROR α). The detailed intracellular signaling of melatonin and/or epigenetic regulation of melatonin is still unknown.

Reviewer #3 (Remarks to the Author):

The authors of this manuscript have assessed the mechanisms by which melatonin suppresses ovarian follicle activation and, over time, delays ovarian aging. The authors show that melatonin can be synthesized by granulosa cells, and expression of the synthetic enzyme SNAT decreases as follicles are activated. They manipulated melatonin exposure by a variety of approaches, including treatment of ovaries with melatonin in vitro and in vivo and use of SNAT knockout mice, all of which show that melatonin can suppress follicle activation and early growth with the associated consequences on ovulation and implantations. Additional experiments yielded results suggesting that melatonin acts by inhibiting the PI3K-AKT-FOXO3 pathway. Long-term treatment with melatonin had no apparent adverse effects on any aspect of reproduction, but did reduce immune cells in plasma.

Although not an entirely novel concept, this study is comprehensive and provides considerable novel results and some initial proof of safety for the use of melatonin to prevent ovarian aging. The authors have addressed the role of melatonin using many experimental approaches, all of which give quite consistent and convincing results. The methods are described in sufficient detail and the results are presented clearly. The manuscript is generally well-written, but some sections will require editorial

revision of grammar. Also, a lot of abbreviations are used without being defined at first use.

Specific comments

1. Lines 28-31. The first sentence is confusing: "a gain-of-function mutation"? Ovarian aging is a natural process by which fertility decreases as follicles become depleted. It does not occur "before they age", but as women age.
2. Line 71-72: The description of the maintenance of the DOR is a little unclear. Is a larger DOR a reflection of a greater PreOR?
3. Line 81: The abbreviations VC and VE are used without definition.
4. Lines 96-98: It is stated the MLT treatment can increase serum LH levels, but sustained elevated gonadotropin levels are indicative of menopause. Can the benefits of MLT treatment in this context be elaborated to describe more fully the improvement in reproduction-related functions?
5. Lines 218-219: A number of abbreviations on these lines are used without definition.
6. Line 222: "which is consistent with previous reports" requires references.
7. Line 246-256: There is quite a bit of redundancy in this section; the repetition could be minimized.
8. Line 267: GTH needs to be defined.
9. Figure 4H: The gene names are barely legible, even when the page is zoomed in.
10. The word "induced" on line 303 and again on line 304 is confusing. I think the authors mean that the genes were differentially expressed, not induced.
11. The SNAT knockout mice seem to be generated specifically for this study. If that is the case, at least a brief description of the phenotype of these mice would be good. There were no notable impacts on animal health or viability? It would be important for the authors to confirm that the knockout resulted in loss of melatonin expression, as expected?
12. Figure 5D: The x-axis label should be modified to reflect age at delivering pups, not children.
13. The source of the data that indicates that Gdf9, Nobox, Figla, Ddx4 and Zp3 gene expression reflect the number of primordial follicles needs to be provided. For example, is it not surprising that Zp3 is expressed in primordial (non-growing) follicles?
14. Line 407: It would be helpful if the physiological concentrations in plasma as previously reported were stated. How does this compare with the dose being administered to the mice? The discussion of this on lines 523-526 is helpful, but the basis for the conclusion that the mouse dosage cannot be applied to humans is not clear. Unless my calculations are incorrect, the dose of 1 mg/kg/mouse per day would be the equivalent of 25-30 micrograms/day for mice that are 25-30 grams. I agree that translation from mouse to human for any drug is far more than just a calculation, but the rationale for suggesting that the dosage could not be achieved in humans is not clear.
15. Lines 417-419: Again, a lot of abbreviations are used without definition.
16. Lines 474-476 state that "we found that MLT-deficient mice did not exhibit premature ovarian failure" – is it not premature ovarian failure that is shown in Figure 5D-F?
17. The overall concept being tested is that endogenous MLT might play a role in suppressing follicle activation and ovarian aging. Is there any evidence that melatonin levels decrease with age, especially after the age of 35?
18. Figure 7: "Delaying" is spelled incorrectly. Also the nature of the connection between the DOR and "cyclic recruitment" is not clear.
19. Some discussion of the relative impacts of melatonin inhibition of both DOR and atresia would be good. If melatonin treatment results in fewer SAFs, but also reduces loss of follicles to atresia, the results overall suggest that the impact on atresia has minimal impact on salvaging follicles in the DOR; i.e. in terms of fertility (ovulations, implantations, pups numbers).

Reviewers' comments:

Reviewer #1 (Remarks to the Author):

This paper provides some evidence that melatonin, either endogenously or exogenously can affect the rate of activation of primordial follicles in the mouse ovary, with potential implications for reproductive lifespan. The authors provide evidence for melatonin synthesis in the granulosa cells of the ovary, pharmacological affects on follicle activation and importantly evidence from a SNAT knockout mouse.

Response: Dear reviewer, we want to begin by thanking for your professional criticisms. In the revised manuscript, we have made a thorough revision according to your criticisms and suggestions.

Specific Comments

1. There are a substantial number of areas of English usage throughout the paper that should be addressed. There are also a number of rather surprising statements, starting with the very first sentence suggesting that ovarian aging is a gain of function mutation. This seems a very strange comment that should be reconsidered. Similarly, on line 34, there is evidence that lots of gonadal hormones triggers cancer but the references cited provide no evidence for this.

Response: Thank you for your professional comments. We apologize for confusing you with the statements in the manuscript. What we want to make clear is that "there could have been germ stem cells in the female gonadal gland, but the natural variation in the evolutionary process causes germ stem cells to fully differentiate into oocytes in the embryo. Thus the loss of oocytes of the female could not be replenished with stem cells, which eventually resulted in ovarian aging. It is apparent that these mutations have been retained as dominant traits during evolution, protecting females from the life-threatening conditions of late pregnancy, while denying them fertility for life ". The above viewpoint is my personal speculation on the cause of ovarian aging, which has not been verified by experiments. In view of this, we deleted the expression "gain of function mutation" in the revised manuscript.

In reply to your comment, we have rewritten Line 34. As "The disorder of reproductive hormones caused by it can trigger many diseases including cardiovascular disease, ovarian cancer, osteoporosis, obesity and menopausal syndrome". "ovarian cancer" is highlighted in that statistics show ovarian cancer is most frequently diagnosed in postmenopausal women ^{1,2}.

2. Line 35. A further similar strange comment is describing ovarian aging as a pacemaker for female body aging.

Response: In the new version, we have changed this statement to "the impact of ovarian aging reaches far beyond just the reproductive system, it also indirectly causes dysfunction in other organs of the body".

3. Line 81. VC and VE presumably referred to the vitamins.

Response: VC and VE refer to vitamin C and vitamin E, respectively. In the revised manuscript, we have made a thorough revision about abbreviations.

4. Figure 1 shows SNAT localization in the ovary. The day 9 and day 17 images show very poor ovarian architecture and should be replaced with better quality fixed tissue. This is also evidenced in the loss of oocytes from the sections that are shown. A wide range of follicles could

also be shown including antral follicles. Negative controls are also not described.

Response: According to your suggestion, we carried out the immunohistochemistry experiment again and demonstrated the figures of negative control. The new images show more clearly the structure of the follicle at various stages (Fig. 1A).

5. There is no analysis of the presence of melatonin receptors within the ovary, which is fundamental to the hypothesis presented.

Response: According to your suggestion, we detected melatonin transmembrane receptors *Mtnr1a* and *Mtnr1b* respectively. The results confirmed that *Mtnr1a*, rather than *Mtnr1b*, was expressed in neonatal mouse ovary (Fig. S2A).

6. The decrease in SNAT mRNA may merely reflect dilution caused by the presence of larger antral follicles and these data would be much more informative if they were expressed relative to another marker of the small follicle population that melatonin is proposed to regulate. This was applied to panel D in figure 1.

Response: Thanks for your professional comments. We also tend to believe that the decline of SNAT mRNA in ovary is not caused by the increase of age, but by the thinning of large follicle. Similarly, after ovary grows, primordial follicles are squeezed into different areas of the ovary cortex by the larger follicles. Therefore, the possibility of concentration gradients of melatonin in different regions of ovary cannot be ruled out at the same time as the overall concentration decreases. We have added the elucidation of the above issues in the discussion section (Line: 632-634).

7. The authors use the term 'extremely significant' repeatedly through the manuscript, this should be changed to just 'significant'.

Response: Thank you for your suggestion. We've changed 'extremely significant' to 'significant'.

8. In figure 2, data on the number of primordial follicles should also be presented as it is possible that the survival of that population might be affected. This is very relevant to interpretation of the number of activated follicles.

Response: Dear reviewer, thank you for your suggestion. Neonatal ovary has a large number of primordial follicles, which make the accurate calculation of them on tissue sections difficult. In addition, we believe that compared with the number of primordial follicles in neonatal ovary, the MLT-driven activation efficiency could hardly have an obvious impact on the total number of primordial follicles in such a short period of time. We hope you will agree with us.

9. Line 209. The authors should provide evidence of lack of activity of the PI3K pathway and changes to localization of foxo3. This does not demonstrate that these pathways are the target of melatonin action on primordial follicles as changes in their activity is a reflection of the changes in follicular activity rather than necessarily showing a direct targeting.

Response: Dear reviewer, we further examined the effect of melatonin deficiency on PI3K-AKT activity in the revised manuscript. The data showed that after *SNAT* was knocked out of mice, the phosphorylation of AKT was enhanced, and MLT administration counteracted the increase in p-AKT level caused by *SNAT*-knockout (Fig. 5F). These results suggested that melatonin indeed inhibited the activity of PI3K-AKT.

With respect to your second question, the regulation mechanism of primordial follicle activation has not yet been clarified, which poses great challenges to the study on the

mechanism of melatonin inhibiting the activation of primordial follicle. At present, it is generally recognized that the PI3K-Akt-FoxO3 pathway controls the activation of primordial follicle. Therefore, when MLT is found to suppress the activation of primordial follicle, what occurs to us is to examine the impact of MLT on PI3K-AKT pathway activity. In fact, it has often been reported that melatonin function by inhibiting the PI3K-AKT pathway. For instance, melatonin alleviates the inflammatory response in the rheumatoid arthritis by inhibiting the PI3K-AKT pathway³; MLT inhibits proliferation, invasion and tumorigenesis of human bladder cancer cell via inhibiting PI3K-Akt-Mtor-MMPs signaling⁴. We strongly agree with your viewpoint that the PI3K-AKT signaling may not be the direct target of melatonin, and melatonin may indirectly affect PI3K-AKT pathway activity through other pathways. But at least, these data indicated that PI3K-AKT-FOXO3 cascade is the one of the targets of melatonin action on primordial follicle activation. In the revised manuscript, we try to verify whether melatonin affects the activation of primordial follicle through Hippo signaling, which has been proved to be responsible for inhibiting the growth and activation of early follicle, and can co-modulate the growth of *in vitro* ovaries through PI3K-AKT pathway⁵⁻⁹. However, our experiment showed that melatonin treatment had no significant effect on the expression of key genes of Hippo signaling pathway in ovary (Fig. S1), and no significant effect on the amount of YAP protein and phosphorylation modification (Fig. 3C).

In the revised version, we also study whether the inhibiting effect of melatonin on follicle activation is mediated by transmembrane receptors. We found that *Mtnr1a* but not *Mtnr1b* was expressed in neonatal mouse ovary, and the constitutive expression level of *Mtnr1a* was not high (Fig. S2A). Later, we add Luzindole, a competitive inhibitor of *Mtnr1a* with 10 times the concentration of melatonin, to the *in vitro* culture system, which still cannot offset the inhibiting effect of melatonin on activation of primordial follicle (Fig. S2B). Therefore, we believe that the effect of melatonin is at least not mediated by its transmembrane receptors. In fact, melatonin is a unique hormone found in animals, plants, bacteria and fungi, contributing to the complexity of melatonin's functions and mechanism of action. The functional versatility and diversity of melatonin has exceeded researchers' expectations, and the most direct evidence is that melatonin has multiple receptor-mediated and receptor-independent actions¹⁰. Studies on its receptors have also confirmed that in addition to the cell membrane receptors *Mtnr1a* and *Mtnr1b*, there are numerous binding sites in the cytosol and in the nucleus, such as QR2, Calmodulin, GPR50, NQO2, RZR/ROR α and VDR, etc.¹¹⁻¹⁴.

To sum up, the unclear activation mechanism of primordial follicle and the complexity of melatonin's mode of action objectively increase the difficulty to reveal the mechanism of melatonin's suppression of primordial follicle activation. Although we have endeavored to improve this study, we can only conclude that MLT reduces the activation of primordial follicle by inhibiting the PI3K-AKT pathway. Whether the inhibiting effect is direct or indirect is not clear yet. As you commented, the reduction of PI3K pathway activity may just be the reflection of melatonin's suppression of follicle activation at the molecular level. We do not shy away from the shortcomings of the research on mechanism, but we hope that this will not obscure the significance of the research itself. Finally, I apologize for not being able to give an accurate answer to your concerns.

10. Line 218. Changes in various serum anti-oxidant markers are demonstrated, without changes in the mRNA activity of the relevant genes in the ovary. The authors should be more cautious as to interpreting this to mean that melatonin improves the anti-oxidant capacity of the ovary.

Response: Thank you for your suggestion. It is indeed not cautious to infer the change of ovary's antioxidant capacity from the increase of serum antioxidant capacity. Therefore, we measured the antioxidant capacity of ovary. The findings showed that MLT significantly increased the activity of total SOD enzyme in ovary. In the revised manuscript, we replaced the previous serum data with data from ovary (Fig. 3E).

11. Figure 4. The histological data are calculated as proportion of different follicle types relative to number of activated follicles. I am not at all convinced by the validity of this analysis and data on the absolute numbers of the different follicle types would be a lot more robust. Similarly the interpretation of the data on the expression levels of FSH and LH receptor is unconvincing, and is difficult to interpret reliably.

Response: I'm sorry for the confusion caused by the way we analyzed the data. In fact, other authors and I have discussed whether proportion or quantity is used to measure the growth of follicle in the analysis of the sections. The final reason why proportion is used as a measure is that it can not only reflect the development status of the follicles, but also directly reflect the distribution of the number of follicles in each stage of the group. According to your suggestion, we use the number of follicle as the measure in the revised manuscript (Fig. 2B,D; Fig. 3A; Fig. 4 A). Although the statistical analysis yielded change of some indicators, the data still show that melatonin inhibits early follicular development and atresia.

As for your second question, there are no clear marker genes of late secondary follicles and small antral follicles. Although *Fshr* and *Lhcgr* are expressed in primordial, primary and early secondary stages, the expression level is relatively low. On the contrary, their expression level increases dramatically in late secondary and small antral follicle, which makes the development of the follicle enter the gonadotrophin-dependent phase. Therefore, we believe that the expressions of *Fshr* and *Lhcgr* in prepubertal mice ovary can directly reflect the number of late secondary (Type5b) and small antral follicles in ovary. We admit that *Fshr* and *Lhcgr* are not perfect marker genes of Type 5b and small antral follicle, but it is reasonable to use the expression levels of *Fshr* and *Lhcgr* as molecular indicators to measure the number of Type 5b and small antral follicle in prepubertal ovary in the case that no marker genes with higher specificity are found. We hope you will agree with us.

12. Figure 4, panel E. The heatmap is inadequately labelled and is essentially not interpretable.

Response: Dear reviewer, we have only kept the volcano map in the revised manuscript to make the picture more concise.

13. Line 371. In addition to follicle numbers, ovarian weights would also be informative here.

Response: Dear reviewer, we also considered weighing the ovaries during sampling, but we gave up this measure because the weight of the ovaries will change with the change of estrus.

14. Line 372. GDF9 and ZP3 are not expressed by oocytes and primordial follicles but rather by oocytes within growing follicles.

Response: Dear reviewer, thanks for your professional criticisms. This problem is caused by a misrepresentation in writing. We know that *GDF9*, *Nobox*, *Figla*, *DDX4* and *ZP3* cannot accurately reflect the number of primordial follicles in ovary. Among them, *Nobox*, *Figla* and *DDX4* are expressed in oocytes of primordial and growing stages, and the purpose of detection of their expression level is to evaluate the total amount of ovary oocytes (including primordial and

growing follicles). Considering that the number of primordial follicles in ovary in the same period is much more than that of growing follicles, we believe that the expression levels of *Nobox*, *Figla* and *DDX4* more reflect the reserve of Primordial follicle. GDF9 and ZP3 are only expressed in growing follicles. The purpose of detecting GDF9 and ZP3 is to further verify the results of histological analysis in Fig.5H at the molecular level, that is, to evaluate the number of growing ovary follicle groups through their expression levels. We have reinterpreted the findings in the revised manuscript (Line: 420-425).

15. Line 440. There is now some discussion of cisplatin out of the blue in this manuscript, which is very confusing as it has not been mentioned earlier in the manuscript. The authors should carefully consider the organisation of the discussion.

Response: Thank you for your suggestion. We've rewritten the discussion.

16. Line 474. I do not think the authors have investigated premature ovarian failure in this study and this comment should be removed.

Response: Thank you for your suggestion. We have deleted this comment from the manuscript.

17. Line 484. Why did the authors choose this specific mouse strain which already has a mutation in melatonin synthesis?

Response: Dear Reviewer, the reason lies in that only *SNAT* knockout mice of the C57BL strain are currently available. Of course, the rationality of using C57BL was evaluated beforehand: (1) The mutations carried by C57BL only lead to lower levels of melatonin in C57BL than other strains, not loss of melatonin, and the effect of melatonin was equally important for C57BL mice. For example, the regulation of rhythmic synthesis of melatonin in pineal gland of C57BL mice and the diurnal variation of melatonin levels under short light stimulation are similar to those of C3H strain (a high melatonin strain). Therefore, C57BL mice remain an important animal model for the study of circadian rhythm¹⁵; (2) *Mtnr1a*-knockout C57BL mice have been produced. The phenotypic data confirmed that knockout of *Mtnr1a* in C57BL mice can produce behavioral rhythm disruption as well as anxiolytic-like activities. This further demonstrates that melatonin plays an important role in the regulation of physiological activities in C57BL mice¹⁶; (3) In mammals, melatonin is mainly synthesized by pineal gland in the nervous system. However, in addition to pineal gland, melatonin can also be locally synthesized in peripheral tissues. Low levels of melatonin in C57BL mice are mainly found in pineal gland, while melatonin levels in peripheral tissues such as blood, thymus, spleen and bone marrow are not much different from those in C3H mice, and even slightly higher in the thymus. These results suggest an extrapineal melatonin synthesis that counterbalance the pineal melatonin deficiency in C57BL mice, and also suggest that melatonin secreted by peripheral tissues and melatonin secreted by pineal gland may be different and independent in function^{17,18}. In view of the above reasons, we believe that it is feasible to select knockout C57BL mice to study how melatonin regulates the physiological function of ovary. And the final experimental results also confirmed *SNAT*-knockout in C57BL mice accelerated the exhaustion of ovarian reserve and age-related fertility decline.

18. Figure 7 has a spelling mistake in 'delaying'.

Response: We have corrected the mistake.

References

1. Yeo W, Ueno T, Lin CH, et al. Treating HR+/HER2- breast cancer in premenopausal Asian women: Asian Breast Cancer Cooperative Group 2019 Consensus and position on ovarian suppression. *Breast Cancer Res Treat.* 2019;177(3):549-559. doi:10.1007/s10549-019-05318-5
2. Ries LAG, Melbert D, Krapcho M, et al., editors. SEER cancer statistics review, 1975–2004. Bethesda, MD: National Cancer Institute; (http://seer.cancer.gov/csr/1975_2004/). Based on November 2006 SEER data submission, posted to the SEER website, 2007)
3. Huang CC, Chiou CH, Liu SC, et al. Melatonin attenuates TNF- α and IL-1 β expression in synovial fibroblasts and diminishes cartilage degradation: Implications for the treatment of rheumatoid arthritis. *J Pineal Res.* 2019;66(3):e12560. doi:10.1111/jpi.12560
4. Chen YT, Huang CR, Chang CL, et al. Jagged2 progressively increased expression from Stage I to III of Bladder Cancer and Melatonin-mediated downregulation of Notch/Jagged2 suppresses the Bladder Tumorigenesis via inhibiting PI3K/AKT/mTOR/MMPs signaling. *Int J Biol Sci.* 2020;16(14):2648-2662. doi:10.7150/ijbs.48358
5. Hsueh AJW, Kawamura K. Hippo signaling disruption and ovarian follicle activation in infertile patients. *Fertil Steril.* 2020;114(3):458-464. doi:10.1016/j.fertnstert.2020.07.031
6. Grosbois J, Demeestere I. Dynamics of PI3K and Hippo signaling pathways during in vitro human follicle activation. *Hum Reprod.* 2018;33(9):1705-1714. doi:10.1093/humrep/dey250
7. Devenuto L, Quintana R, Quintana T. In vitro activation of ovarian cortex and autologous transplantation: A novel approach to primary ovarian insufficiency and diminished ovarian reserve. *Hum Reprod Open.* 2020; 2020(4):hoaa046. doi:10.1093/hropen/hoaa046
8. Kawamura K, Cheng Y, Suzuki N, et al. Hippo signaling disruption and Akt stimulation of ovarian follicles for infertility treatment. *Proc Natl Acad Sci U S A.* 2013;110(43):17474-17479. doi:10.1073/pnas.1312830110
9. Lo Sardo F, Muti P, Blandino G, Strano S. Melatonin and Hippo Pathway: Is There Existing Cross-Talk?. *Int J Mol Sci.* 2017;18(9):1913. Published 2017 Sep 6. doi:10.3390/ijms18091913
10. Reiter RJ, Tan DX, Manchester LC, Pilar Terron M, Flores LJ, Koppisepi S. Medical implications of melatonin: receptor-mediated and receptor-independent actions. *Adv Med Sci.* 2007;52:11-28.
11. Boutin JA, Ferry G. Is There Sufficient Evidence that the Melatonin Binding Site MT3 Is Quinone Reductase 2?. *J Pharmacol Exp Ther.* 2019;368(1):59-65. doi:10.1124/jpet.118.253260
12. Benítez-King G, Huerto-Delgado L, Antón-Tay F. Binding of 3H-melatonin to calmodulin. *Life Sci.* 1993;53(3):201-207. doi:10.1016/0024-3205(93)90670-x
13. Emet M, Ozcan H, Ozel L, Yayla M, Halici Z, Hacimuftuoglu A. A Review of Melatonin, Its Receptors and Drugs. *Eurasian J Med.* 2016;48(2):135-141. doi:10.5152/eurasianjmed.2015.0267
14. Fang N, Hu C, Sun W, et al. Identification of a novel melatonin-binding nuclear receptor: Vitamin D receptor. *J Pineal Res.* 2020;68(1):e12618. doi:10.1111/jpi.12618
15. von Gall C, Lewy A, Schomerus C, et al. Transcription factor dynamics and neuroendocrine signalling in the mouse pineal gland: a comparative analysis of melatonin-deficient C57BL mice and melatonin-proficient C3H mice. *Eur J Neurosci.* 2000;12(3):964-972. doi:10.1046/j.1460-9568.2000.00990.x
16. Adamah-Biassi EB, Hudson RL, Dubocovich ML. Genetic deletion of MT1 melatonin receptors alters spontaneous behavioral rhythms in male and female C57BL/6 mice. *Horm Behav.* 2014;66(4):619-627. doi:10.1016/j.yhbeh.2014.08.012
17. Gómez-Corvera A, Cerrillo I, Molinero P, et al. Evidence of immune system melatonin production by two pineal melatonin deficient mice, C57BL/6 and Swiss strains. *J Pineal Res.* 2009;47(1):15-22. doi:10.1111/j.1600-079X.2009.00683.x
18. Sanchez-Hidalgo M, de la Lastra CA, Carrascosa-Salmoral MP, et al. Age-related changes in melatonin synthesis in rat extrapineal tissues. *Exp Gerontol.* 2009;44(5):328-334. doi:10.1016/j.exger.2009.02.002

Reviewer #2 (Remarks to the Author):

Review of the manuscript COMMSBIO-20-2920, entitled "Melatonin as an endogenous hormone to slow down the exhaustion of ovarian follicle reserve in mice—a novel insight into its roles in early folliculogenesis and ovarian aging."

Overview and general comments

The authors investigated the role of melatonin on ovarian aging and the mechanism of melatonin to delay ovarian aging. In vitro and in vivo experiments showed that melatonin inhibited primordial follicle activation by inhibiting the PI3K-AKT-FOXO3 pathway, and melatonin also inhibits early follicle growth and atresia. They also confirmed that SNAT knockout in mice resulted in a significant increase in follicle activation and atresia, and eventually accelerated ovarian aging. They concluded that endogenous melatonin is involved in the regulation of ovarian aging, and reveals that melatonin delays ovarian aging by inhibiting primordial follicle activation, early follicle growth and atresia.

This study is variable to understand the role of melatonin on ovarian physiology and ovarian aging. They demonstrated that not only direct anti-oxidative effect of melatonin but also novel mechanism of melatonin to inhibit primordial follicle activation are involved in the regulation of ovarian aging. The study design is well refined.

Major comments

1. Melatonin produced by pineal gland is secreted into the blood circulation, however, extra-pineal melatonin produced by other organs stay inside the cell and protects themselves from oxidative stress. If primordial and early follicles produce melatonin, how melatonin inhibits primordial activation? Autocrine action? Paracrine action? Does melatonin inhibit primordial follicle activation themselves or other primordial follicles? How many animals per group in each experiment?

2. Melatonin inhibits primordial follicle activation by suppressing PI3K-AKT-FOXO3 pathway. It may through melatonin membrane receptor (MT1, MT2) or melatonin nuclear receptor (ROR α). The detailed intracellular signaling of melatonin and/or epigenetic regulation of melatonin is still unknown.

Response: Dear reviewer, we want to begin by thanking for your professional criticisms.

We tend to hold that melatonin inhibits activation of primordial follicle through both autocrine and paracrine ways. According to previous reports, He *et al.* have confirmed that there is a paracrine regulatory pathway of melatonin in follicle, that is, melatonin secreted by follicular cells during ovulation acts on neighboring granulosa cells and then promotes the differentiation of granulosa cells into luteal cells ¹. Yang *et al.* found that early embryos can synthesize melatonin through mitochondria. Mitochondrial synthesis of MLT plays an important role in maintaining embryonic mitochondrial function and reducing oxidative damage. This reflects the autocrine regulation of melatonin in the reproductive system ². In 2017, Suofu Y *et al.* discovered a novel secretion pattern of melatonin at the organelle level. They identified Mtnr1a receptors on the outer membrane of mitochondria of nerve cells, and demonstrated that melatonin synthesized by mitochondria can directly bind to Mtnr1a receptors on the outer membrane to regulate the process of cell apoptosis. They therefore proposed a new term, "automitocrine," analogous to

"autocrine" when a similar phenomenon occurs at the cellular level, to describe this unexpected intracellular organelle ligand-receptor pathway³. The above studies indicate that melatonin can exert physiological functions in both paracrine and autocrine ways, or even both ways occur simultaneously. Our IHC data shows that SNAT exists in all stages of follicles, and melatonin secreted by these follicles will gather in the ovary tissues without difference, and then affect the activation, development or atresia of the follicle. In this case, it is difficult to distinguish between paracrine and autocrine effects. We tend to believe that melatonin inhibits activation of primordial follicles in both autocrine and paracrine ways. We hope you will agree with us. In the revised manuscript, we labeled the number of animals used in the caption.

As for your second question, we study whether the inhibiting effect of melatonin on follicle activation is mediated by its transmembrane receptors. We found that *Mtnr1a* but not *Mtnr1b* was expressed in neonatal mouse ovary, and the constitutive expression level of *Mtnr1a* was not high (Fig. S2A). Later, we add Luzindole, a competitive inhibitor of *Mtnr1a* with 10 times the concentration of melatonin, to the *in vitro* culture system, which still cannot offset the inhibiting effect of melatonin on activation of primordial follicle (Fig. S2B). Therefore, we believe that the effect of melatonin is at least not mediated by its transmembrane receptors. In fact, melatonin is a unique hormone found in animals, plants, bacteria and fungi, contributing to the complexity of melatonin's functions and mechanism of action. The functional versatility and diversity of melatonin has exceeded researchers' expectations, and the most direct evidence is that melatonin has multiple receptor-mediated and receptor-independent actions⁴. Studies on its receptors have also confirmed that in addition to the cell membrane receptors *Mtnr1a* and *Mtnr1b*, there are numerous binding sites in the cytosol and in the nucleus, such as QR2, Calmodulin, GPR50, NQO2, RZR/ROR α and VDR, etc.⁵⁻⁸. The above characteristics of melatonin pose challenges to study its specific mechanism of action. Although we have tried our best to conduct a series of additional experiments, we are sorry that we are not able to give you an accurate answer to your concerns.

References

1. He C, Ma T, Shi J, et al. Melatonin and its receptor MT1 are involved in the downstream reaction to luteinizing hormone and participate in the regulation of luteinization in different species. *J Pineal Res.* 2016;61(3):279-290. doi:10.1111/jpi.12345
2. Yang M, Tao J, Wu H, et al. Aanat Knockdown and Melatonin Supplementation in Embryo Development: Involvement of Mitochondrial Function and DNA Methylation. *Antioxid Redox Signal.* 2019;30(18):2050-2065. doi:10.1089/ars.2018.7555
3. Suofu Y, Li W, Jean-Alphonse FG, et al. Dual role of mitochondria in producing melatonin and driving GPCR signaling to block cytochrome c release. *Proc Natl Acad Sci U S A.* 2017;114(38):E7997-E8006. doi:10.1073/pnas.1705768114
4. Reiter RJ, Tan DX, Manchester LC, Pilar Terron M, Flores LJ, Koppisepi S. Medical implications of melatonin: receptor-mediated and receptor-independent actions. *Adv Med Sci.* 2007;52:11-28.
5. Boutin JA, Ferry G. Is There Sufficient Evidence that the Melatonin Binding Site MT3 Is Quinone Reductase 2?. *J Pharmacol Exp Ther.* 2019;368(1):59-65. doi:10.1124/jpet.118.253260
6. Benítez-King G, Huerto-Delgadillo L, Antón-Tay F. Binding of 3H-melatonin to calmodulin. *Life Sci.* 1993;53(3):201-207. doi:10.1016/0024-3205(93)90670-x
7. Emet M, Ozcan H, Ozel L, Yayla M, Halici Z, Hacimuftuoglu A. A Review of Melatonin, Its Receptors and Drugs. *Eurasian J Med.* 2016;48(2):135-141. doi:10.5152/eurasianjmed.2015.0267
8. Fang N, Hu C, Sun W, et al. Identification of a novel melatonin-binding nuclear receptor: Vitamin D receptor. *J Pineal Res.* 2020;68(1):e12618. doi:10.1111/jpi.12618

Reviewer #3 (Remarks to the Author):

The authors of this manuscript have assessed the mechanisms by which melatonin suppresses ovarian follicle activation and, over time, delays ovarian aging. The authors show that melatonin can be synthesized by granulosa cells, and expression of the synthetic enzyme SNAT decreases as follicles are activated. They manipulated melatonin exposure by a variety of approaches, including treatment of ovaries with melatonin in vitro and in vivo and use of SNAT knockout mice, all of which show that melatonin can suppress follicle activation and early growth with the associated consequences on ovulation and implantations. Additional experiments yielded results suggesting that melatonin acts by inhibiting the PI3K-AKT-FOXO3 pathway. Long-term treatment with melatonin had no apparent adverse effects on any aspect of reproduction, but did reduce immune cells in plasma.

Although not an entirely novel concept, this study is comprehensive and provides considerable novel results and some initial proof of safety for the use of melatonin to prevent ovarian aging. The authors have addressed the role of melatonin using many experimental approaches, all of which give quite consistent and convincing results. The methods are described in sufficient detail and the results are presented clearly. The manuscript is generally well-written, but some sections will require editorial revision of grammar. Also, a lot of abbreviations are used without being defined at first use.

Response: Dear reviewer, we want to begin by thanking you for your professional criticisms and suggestions. Based on your comments, we have made the following revisions to the manuscript:

Specific comments

1. Lines 28-31. The first sentence is confusing: "a gain-of-function mutation"? Ovarian aging is a natural process by which fertility decreases as follicles become depleted. It does not occur "before they age", but as women age.

Response: We apologize for confusing you with the statements in the manuscript. What we want to make clear is that "there could have been germ stem cells in the female gonadal gland, but the natural variation in the evolutionary process causes germ stem cells to fully differentiate into oocytes in the embryo. Thus the loss of oocytes of the female could not be supplemented with stem cells, which eventually resulted in ovarian aging. It is apparent that these mutations have been retained as dominant traits during evolution, protecting females from the life-threatening conditions of late pregnancy, while denying them fertility for life ". The above viewpoint is my personal speculation on the cause of ovarian aging, which has not been verified by experiments. In view of this, we deleted the expression "gain of function mutation" in the revised manuscript. In addition, we change "before age" to "as age". Thank you for your professional comments.

2. Line 71-72: The description of the maintenance of the DOR is a little unclear. Is a larger DOR a reflection of a greater PreOR?

Response: Dear reviewer, DOR and PreOR are like two interconnected reservoirs with huge differences in volume. On the one hand, the constant loss of early growing follicles in DOR happens through atresia and cyclic recruitment. On the other hand, the activation of PreOR follicle can continuously replenish the loss of DOR, so that DOR is in a dynamic balance, which is determined by the replenishing ability of PreOR. From the birth of the animal to the whole

period with strong fertility, the reserve of primordial follicles in ovary is much higher than the number of growing follicles, so PreOR has enough capacity to replenish the loss of DOR. The scale of DOR during this period was relatively stable; In the reproductive decline period, primordial follicle's reserve is gradually reduced, and PreOR's ability to replenish DOR begins to decline. So in this period, the scale of DOR gradually decreases, and eventually diminishes when the reserve of primordial follicle decreases to a minimum. Therefore, under normal circumstances, we believe that a larger DOR at most reflects the high activation efficiency of primordial follicle in PreOR, but it's inadequate to indicate that PreOR is necessarily larger. But in the reproductive decline period, a smaller DOR reflects the lower reserve of primordial follicles in PreOR.

3. Line 81: The abbreviations VC and VE are used without definition.

Response: VC and VE refer to vitamin C and vitamin E, respectively. In the revised manuscript, we have made a thorough revision about abbreviations.

4. Lines 96-98: It is stated the MLT treatment can increase serum LH levels, but sustained elevated gonadotropin levels are indicative of menopause. Can the benefits of MLT treatment in this context be elaborated to describe more fully the improvement in reproduction-related functions?

Response: I'm very sorry for the wrong interpretation. In this paper, it was reported that nocturnal melatonin level decreased gradually with age. The authors suggested that nocturnal melatonin level decrease might play a role in the occurrence and progression of menopause. A 6-month double-blind study of melatonin intake confirmed a significant decrease of LH levels in premenopausal women and a significant decrease of FSH levels in all postmenopausal women. Thank you for pointing out this mistake and we have corrected it in the revised manuscript.

5. Lines 218-219: A number of abbreviations on these lines are used without definition.

Response: In the revised manuscript, we have made a thorough revision about abbreviations.

6. Line 222: "which is consistent with previous reports" requires references.

Response: Thank you for your suggestion. The reference has been cited in the revised manuscript.

7. Line 246-256: There is quite a bit of redundancy in this section; the repetition could be minimized.

Response: Thank you for your suggestion. We have simplified the description.

8. Line 267: GTH needs to be defined.

Response: In the revised version we provide the full name of GTH (gonadotropins).

9. Figure 4H: The gene names are barely legible, even when the page is zoomed in.

Response: The font size in figure 4H has been enlarged

10. The word "induced" on line 303 and again on line 304 is confusing. I think the authors mean that the genes were differentially expressed, not induced.

Response: Thank you for your suggestion. We have changed "significantly induced" to "differentially expressed".

11. The SNAT knockout mice seem to be generated specifically for this study. If that is the case, at least a brief description of the phenotype of these mice would be good. There were no notable impacts on animal health or viability? It would be important for the authors to confirm that the

knockout resulted in loss of melatonin expression, as expected?

Response: Dear reviewer, thank you for your suggestion. We have supplemented the relevant experiments. It was confirmed that MLT in mice could not be detected after *SNAT* was knocked out of mice (Fig. 5B). *SNAT* knockout had no significant effect on the growth of mice (Fig. S4A). However, it is worth noting that *SNAT* knockout resulted in a significant decrease in spleen weight (Fig. S4B) and a significant decrease in the number of white blood cells (Fig. S4C). It suggests that MLT performs an important role in immune regulation.

12. Figure 5D: The x-axis label should be modified to reflect age at delivering pups, not children.

Response: Thank you for your suggestion. We have corrected the mistake.

13. The source of the data the indicates that *Gdf9*, *Nobox*, *Figla*, *Ddx4* and *Zp3* gene expression reflect the number of primordial follicles needs to be provided. For example, is it not surprising that *Zp3* is expressed in primordial (non-growing) follicles?

Response: Dear reviewer, thanks for your professional criticisms. This problem is caused by a misrepresentation in writing. We are clear that *GDF9*, *Nobox*, *Figla*, *DDX4* and *ZP3* cannot accurately reflect the number of primordial follicles in ovary. Among them, *Nobox*, *Figla* and *DDX4* are expressed in oocytes at the stage of primordial and growing follicle, and the purpose of detecting the expression level is to evaluate the total amount of ovary oocytes (primordial + growing). Considering that the number of primordial follicles in ovary in the same period is much more than that of growing follicles, we believe that the expression levels of *Nobox*, *Figla* and *DDX4* more reflect the reserve of primordial follicle. *GDF9* and *ZP3* are only expressed in growing follicles. The purpose of detecting *GDF9* and *ZP3* is to further verify the results of histological analysis in Fig.5E at the molecular level, that is, to evaluate the number of growing ovary follicle groups through their expression levels. We have reinterpreted the results in a revised manuscript. The original data of related gene quantification is also enclosed in the source data (Supplementary data-1).

14. Line 407: It would be helpful if the physiological concentrations in plasma as previously reported were stated. How does this compare with the dose being administered to the mice? The discussion of this on lines 523-526 is helpful, but the basis for the conclusion that the mouse dosage cannot be applied to humans is not clear. Unless my calculations are incorrect, the dose of 1 mg/kg/mouse per day would be the equivalent of 25-30 micrograms/day for mice that are 25-30 grams. I agree that translation from mouse to human for any drug is far more than just a calculation, but the rationale for suggesting that the dosage could not be achieved in humans is not clear.

Response: Dear reviewer, your calculation is correct. At our dose, the daily melatonin intake of a 30-gram rat is about 30 micrograms. If the same dose is translated into human consumption, the average 50kg woman consumes about 50mg per day. However, we checked that the highest recommended intake for melatonin supplements on the market is 10mg per day (see the picture below), which is lower than our translation of 50mg. Therefore, we should be cautious in our discussion to say that this experimental dose should not be applied directly to humans. However, as a widely-used health product, the safety of melatonin can stand the test. Meanwhile, considering that the basic metabolism of mice is stronger than that of humans, we are optimistic that it is possible to achieve similar effects with mice at a lower dose in humans.

15. Lines 417-419: Again, a lot of abbreviations are used without definition.

Response: In the revised version we provide the full names.

16. Lines 474-476 state that “we found that MLT-deficient mice did not exhibit premature ovarian failure” – is it not premature ovarian failure that is shown in Figure 5D-F?

Response: Dear reviewer, we believe that premature ovary failure refers to the complete loss of fertility in the normal reproductive years of the animal. Therefore, we believe that the data in Figure 5D-F do not represent premature ovary failure in mice. To avoid misunderstanding, this sentence has been deleted from the revised version.

17. The overall concept being tested is that endogenous MLT might play a role in suppressing follicle activation and ovarian aging. Is there any evidence that melatonin levels decrease with age, especially after the age of 35?

Response: Dear reviewer, melatonin levels vary throughout human life and are known to decrease with age. In women, this age-related decrease in melatonin levels coincides with the menopause²⁻⁴. In addition, Tong *et al.*, assessed the levels of MLT in human follicular fluid and found good correlations between melatonin levels with age, anti-Müllerian hormone (AMH) and baseline follicle-stimulating hormone (bFSH), all of which have been used to predict ovarian reserve. They proposed that MLT levels in follicular fluid may serve as a biochemical marker for IVF outcomes and predicting ovarian reserve⁵.

18. Figure 7: “Delaying” is spelled incorrectly. Also the nature of the connection between the DOR and “cyclic recruitment” is not clear.

Response: We corrected the spelling error. The small antral follicles in DOR have two possibilities. If there is no sufficient gonadotropins stimulation, the small antral follicles will be eliminated by atresia. Gonadotropins, on the other hand, will save those follicles from atresia, allowing them to eventually develop into mature follicles and ovulate. In normal estrus cycle, female gonadotropins are secreted periodically, which only make follicles periodically induced to continuously develop. Such process is the "cyclic recruitment".

19. Some discussion of the relative impacts of melatonin inhibition of both DOR and atresia would be good. If melatonin treatment results in fewer SAFs, but also reduces loss of follicles to atresia, the results overall suggest that the impact on atresia has minimal impact on salvaging

follicles in the DOR; i.e. in terms of fertility (ovulations, implantations, pups numbers).

Response: Thank you for your suggestion. In the new version, we have revised the discussion. Follicles in DOR either recruit periodically with the help of gonadotropins, or lead to atresia and apoptosis, but they don't remain at rest for a long time. For prepubertal mice whose hypothalamic pituitary-ovarian axis has not yet been established, the follicles in DOR cannot be recruited periodically without sufficient GTH, but are cleared in an atresia manner. Therefore, without ruling out the direct inhibition of follicular atresia by MLT, we tend to believe that the reduction of the number of follicles in atresia is indirectly caused by the reduction of DOR size by MLT.

References

1. Bellipanni G, Bianchi P, Pierpaoli W, Bulian D, Ilyia E. Effects of melatonin in perimenopausal and menopausal women: a randomized and placebo controlled study. *Exp Gerontol.* 2001;36(2):297-310. doi:10.1016/s0531-5565(00)00217-5
2. Gursoy AY, Kiseli M, Caglar GS. Melatonin in aging women. *Climacteric.* 2015;18(6):790-796. doi:10.3109/13697137.2015.1052393
3. Reiter RJ. Pineal function during aging: attenuation of the melatonin rhythm and its neurobiological consequences. *Acta Neurobiol Exp (Wars).* 1994;54 Suppl:31-39.
4. McCully KS. Communication: Melatonin, Hyperhomocysteinemia, Thioretinaco Ozonide, Adenosylmethionine and Mitochondrial Dysfunction in Aging and Dementia. *Ann Clin Lab Sci.* 2018;48(1):126-131.
5. Tong J, Sheng S, Sun Y, et al. Melatonin levels in follicular fluid as markers for IVF outcomes and predicting ovarian reserve. *Reproduction.* 2017;153(4):443-451. doi:10.1530/REP-16-0641

REVIEWERS' COMMENTS:

Reviewer #1 (Remarks to the Author):

The ms is much improved, though there are English-usage issues throughout. I have the following relatively minor comments:

L18 'several immunological indexes' is not meaningful, please adapt

L27 the first sentence does not make any sense, how is this related to avoiding issues related to elderly pregnancy? Please reconsider this sentence.

L113 SNAT needs explanation, not previously mentioned

Fig 3E: useful new data, but better presented with separate left and right Y axes as the concs of the 2 products are so different

Reviewer #2 (Remarks to the Author):

Review of the manuscript COMMSBIO-20-2920R1, entitled " Melatonin as an endogenous hormone to slow down the exhaustion of ovarian follicle reserve in mice—a novel insight into its roles in early folliculogenesis and ovarian aging."

The authors revised this article according to reviewer's suggestions. The manuscript has been much improved.

Reviewer #3 (Remarks to the Author):

The authors have satisfactorily addressed all of my previous comments and concerns, and have provided additional and supportive data. Thank you. I have no further concerns.

Reviewer #1 (Remarks to the Author):

The ms is much improved, though there are English-usage issues throughout. I have the following relatively minor comments:

1. L18 'several immunological indexes' is not meaningful, please adapt.

Response: Dear reviewers, we have condensed the abstract and this statement has been removed.

2. L27 the first sentence does not make any sense, how is this related to avoiding issues related to elderly pregnancy? Please reconsider this sentence.

Response: Thank you for your comment, we have changed this sentence to "The exhaustion of ovarian follicle reserve leads to the loss of ovarian function in middle-aged females, namely ovarian aging".

3. L113 SNAT needs explanation, not previously mentioned

Response: Thank you for your suggestion, we have defined SNAT in detail in the new version.

4. Fig 3E: useful new data, but better presented with separate left and right Y axes as the concs of the 2 products are so different

Response: Thank you very much for pointing out the issue, we have redrawn this figure according to your suggestion.